# CRISPR/Cas9 recombineering-mediated deep mutational scanning of essential genes in *Escherichia coli*

Alaksh Choudhury[1,2], Jacob A Fenster[1,†], Reilly G Fankhauser[3,†], Joel L Kaar[1], Olivier Tenaillon[2,‡,*] & Ryan T Gill[1,3,4,‡,**] iD

## Abstract

**Deep mutational scanning can provide significant insights into the function of essential genes in bacteria. Here, we developed a high-throughput method for mutating essential genes of *Escherichia coli* in their native genetic context. We used Cas9-mediated recombineering to introduce a library of mutations, created by error-prone PCR, within a gene fragment on the genome using a single gRNA pre-validated for high efficiency. Tracking mutation frequency through deep sequencing revealed biases in the position and the number of the introduced mutations. We overcame these biases by increasing the homology arm length and blocking mismatch repair to achieve a mutation efficiency of 85% for non-essential genes and 55% for essential genes. These experiments also improved our understanding of poorly characterized recombineering process using dsDNA donors with single nucleotide changes. Finally, we applied our technology to target *rpoB*, the beta subunit of RNA polymerase, to study resistance against rifampicin. In a single experiment, we validate multiple biochemical and clinical observations made in the previous decades and provide insights into resistance compensation with the study of double mutants.**

**Keywords** CRISPR-Cas9; deep mutational scanning; essential genes; genome editing; recombineering

**Subject Categories** Methods & Resources; Biotechnology & Synthetic Biology

**Mol Syst Biol. (2020) 16: e9265**

## Introduction

The function of proteins, which drive all cellular processes, can be uncovered by studying mutations in their sequence. Historically, the analysis of the impact of a handful of mutations on protein stability and activity helped lay the foundations of protein science (Fersht *et al*, 1985; Wells, 1990; Clackson & Wells, 1995; Bloom *et al*, 2006; Romero & Arnold, 2009). More recently, the advent of next-generation DNA sequencing platforms has paved way for the emergence of high-throughput sequence-to-function mapping such as deep mutational scanning (DMS), which expands our access to previously unexplored areas of fitness landscapes for proteins (Araya & Fowler, 2011; Fowler & Fields, 2014). In DMS, activities of thousands of mutants in large libraries, covering most or potentially all possible substitutions in the protein sequence, are measured simultaneously using next-generation sequencing (Araya & Fowler, 2011; Fowler & Fields, 2014). This more complete picture of the protein sequence space has the potential to provide unprecedented mechanistic insights into protein structure and activity, evolution, epistasis (Sarkisyan *et al*, 2016; Kemble *et al*, 2019), intracellular behavior, and disease phenotypes (Araya & Fowler, 2011; Fowler & Fields, 2014).

Essential bacterial genes are an especially interesting target for DMS because they play an important role in bacterial evolution (Long *et al*, 2015; Maddamsetti *et al*, 2017), the emergence of antibiotic resistance (Walsh, 2000; Allen *et al*, 2010), and strain engineering (Winkler *et al*, 2016; de Jong *et al*, 2017). It is important to modify the essential genes in their native genomic context. Expressing essential genes on plasmids alters the cellular fitness because of different expression levels due to copy number effects (Gibson *et al*, 2013) and the loss of epigenetic regulation. Current genome mutagenesis techniques suffer from low-editing efficiencies and mutational biasing, which greatly decrease the quality of the fitness data (i.e., due to the overabundance of wild type or a few overrepresented members). As such, comprehensive DMS of essential genes using these approaches has remained elusive, especially in bacteria.

Bacterial genome-editing technologies have advanced greatly in the past decade. One technology is multiplexed automated genome

1   Department of Chemical and Biological Engineering, University of Colorado, Boulder, CO, USA
2   IAME, INSERM, Université de Paris, Paris, France
3   Renewable & Sustainable Energy Institute, University of Colorado, Boulder, CO, USA
4   Novo Nordisk Foundation Center for Biosustainability, Danish Technical University, Copenhagen, Denmark
    *Corresponding author. Tel: +33 1 57 27 77 45; E-mail: olivier.tenaillon@inserm.fr
    **Corresponding author. Tel: +45 93 51 19 29; E-mail: rtg@biosustain.dtu.dk
    †These authors contributed equally to this work
    ‡These authors contributed equally to this work

engineering (MAGE) that is based on lambda Red-mediated recombination of single-stranded oligos for desired allelic exchange on the genome or "Recombineering" (Wang et al, 2009). The efficiency of introducing a single nucleotide change using recombineering is often very low and context-dependent (Sharan et al, 2009). Using MAGE, efficiency and throughput are improved by re-transforming the same large pool of oligos over repeated cycles (Wang et al, 2009). This technique has been successfully used for several outstanding synthetic biology and metabolic engineering applications (Wang et al, 2009; Sandoval et al, 2012; Lajoie et al, 2013b; Raman et al, 2014; Amiram et al, 2015). However, it may be difficult to apply MAGE for DMS of essential genes. Due to the repeated transformation cycles during library construction, the mutations that are deleterious or even neutral to the host would be lost (Wang et al, 2009). Repeated heat shock and electroporation during the recombineering cycles in the MAGE protocol (Wang et al, 2009) would also add additional stress that would be detrimental to the diversity in essential genes. Additionally, in order to increase the efficiency of recombination in MAGE, significant modifications such as deletion of methyl-directed mismatch repair (MMR) and DNA primase (dnaG) are required, which change the native genetic context (Wang et al, 2009; Sawitzke et al, 2011). Deletion of mismatch repair systems increases the background mutation rate (Isaacs et al, 2011), which may also confound fitness estimates.

Genome-editing technologies developed using CRISPR/Cas9-mediated recombineering have helped address several challenges with MAGE. A chimeric guide RNA (gRNA) programs the Cas9 endonuclease to induce a DNA double-strand break (DSB) at any genomic target upstream of a 5′-NGG-3′ PAM sequence and complementary to the 20-bp spacer sequence in the gRNA (Jinek et al, 2012). In several bacteria, including Escherichia coli, the Cas9:gRNA-mediated DNA DSB induces cell death due to a lack of adequate DSB repair pathways (Jiang et al, 2013). Therefore, Cas9:gRNA-induced DSBs can select for PAM substitutions introduced by recombineering (Cong et al, 2013; Jiang et al, 2013). Synonymous PAM-inactivating mutations (SPMs) can be coupled to other mutations in the same recombination template for precise genome manipulation with high efficiency using a single transformation step (Pyne et al, 2015; Reisch & Prather, 2015; Bassalo et al, 2016; Chung et al, 2017; Wang et al, 2018).

High-throughput genome editing with Cas9-mediated recombineering was achieved recently using CRISPR-enabled trackable genome engineering (CREATE) (Garst et al, 2017). Using CREATE, the DNA encoding the gRNA expressed under a constitutive promoter was covalently linked to the DNA repair template on 250-bp editing cassettes (Garst et al, 2017). Over 100,000 editing cassettes can be synthesized on microarray chips and subsequently cloned in high-throughput into cells with active Cas9 and lambda Red recombination to generate genome-wide mutation libraries. The editing cassettes on the plasmid also serve as the barcode to track the mutations before and after selection to assign fitness scores to each mutation (Garst et al, 2017). The technology has been used for directed evolution of E. coli proteins, pathways, and strains (Shalem et al, 2014; Cobb et al, 2015; Cho et al, 2017; Liang et al, 2017; Liu et al, 2017; Lu et al, 2017; Wu et al, 2017a,b; Zhu et al, 2017; Bassalo et al, 2018). However, applying CREATE for DMS proved to be challenging. Anywhere between 10 and 60% of randomly chosen gRNA targeting different genomic loci have been shown not to induce Cas9:gRNA-induced cell death (Cui & Bikard, 2016; Zerbini et al, 2017). Consequently, due to variable selection, editing efficiency can vary between 0 and 100% across gRNAs (Garst et al, 2017; Zerbini et al, 2017). Cells with gRNAs that fail to induce DSB-mediated cell death can grow significantly faster than cells with active gRNAs undergoing DSBs and editing (Jiang et al, 2015; Cui & Bikard, 2016). Consequently, in high-throughput non-DSB-inducing gRNAs, with low-editing efficiency, take over the population and reduce overall editing efficiency to only ~ 1–4% (Bassalo et al, 2018). Several gRNAs also cause unintended mutations on the genome that are not encoded in the repair template (Cui & Bikard, 2016; Zerbini et al, 2017). Consequently, cells with no edits and unintended mutations can be falsely tracked as beneficial mutations. Finally, each gRNA is coupled to a different synonymous PAM mutation (SPM) and synonymous mutations can lead to significant fitness effects, especially in essential genes (Lind et al, 2010; Agashe et al, 2013; Lajoie et al, 2013a). Because of these limitations, CREATE experiments have largely been limited to finding mutants with large fitness effects in the presence of strong selective pressures (Bassalo et al, 2018; Pines et al, 2018).

We posited that in order to target a single genomic locus, we could use a single pre-screened gRNA and synonymous PAM-inactivating mutations (SPM). In this study, we discuss the CRISPR/Cas9-mediated genomic error-prone editing (CREPE) technology. As opposed to other Cas9-mediated high-throughput technologies in E. coli, in the CREPE protocol we use a single gRNA to integrate an error-prone PCR library of the target with the SPM on the genome (Fig 1). Recently, a similar technology, CASPER, was reported in yeast (Jakočiūnas et al, 2018). However, yeast has a significantly higher recombination efficiency than bacteria such as E. coli. Recombination efficiency with linear dsDNA templates is very low in E. coli (Murphy et al, 2000), and recombineering using dsDNA template with limited single nucleotide changes is poorly understood. Therefore, we varied the homology arm length and the Cas9 recombineering system to improve recombination and our understanding of recombination using a repair template with single nucleotide changes. We successfully developed a platform that efficiently generates unbiased and diverse genomic mutant libraries with > 80% editing efficiency for non-essential genes and > 55% efficiency for essential genes. Additionally, while CASPER was used for directed evolution, we adapted CREPE for use as a DMS platform to study essential E. coli genes in their native genomic context. Using CREPE, we scored the fitness of naturally accessible mutations in the RNA polymerase beta subunit that confer resistance to rifampicin.

## Results

### CREPE protocol

In the CREPE workflow (Fig 1), we initially screened for a gRNA centered around the genomic target of interest that enables over 95% editing efficiency for replacing the NGG PAM with the synonymous PAM mutation (SPM) to be used in the repair template. We also ensured that the SPM does not affect the fitness of the cells and that their growth is comparable to wild-type E. coli. In order to use a single gRNA to incorporate multiple mutations, we link the synonymous PAM mutation and secondary targeted mutations on

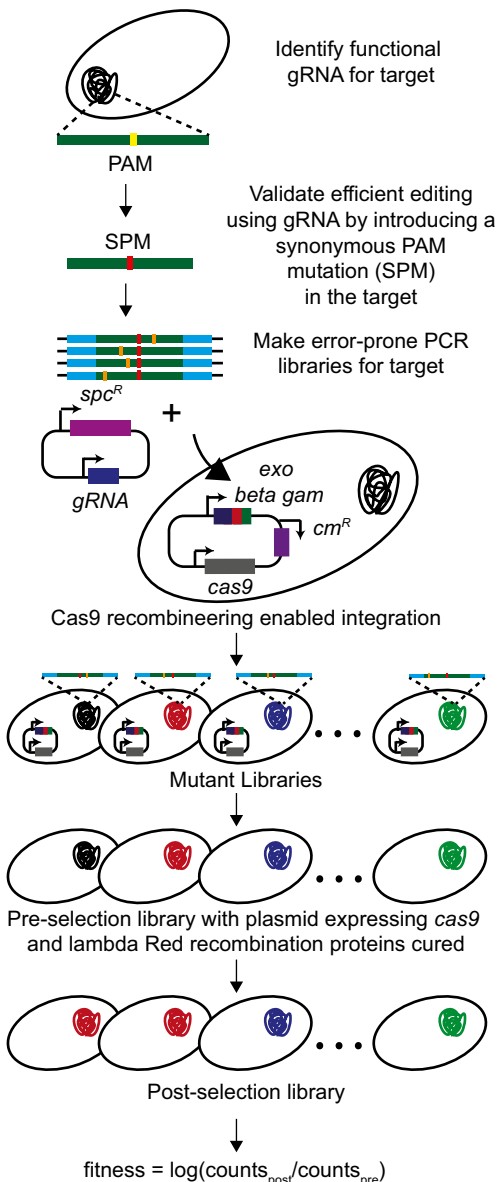

**Figure 1.  CREPE strategy.**
General workflow for the CREPE protocol.

the same repair template. Since we wanted to target as large a window as possible (300–400 bp long) for mutagenesis using a single gRNA, a dsDNA repair template as opposed to single-stranded oligos to overcome the length limitations of oligo synthesis was used. We amplified and cloned the target region with the SPM and sufficient unmutated end homology, which was used for recombination, into a plasmid and develop error-prone PCR libraries (Fig 1). We amplified and co-transformed the linear donor error-prone PCR library with the gRNA-encoding plasmid in cells with active Cas9 and lambda Red recombination proteins, encoded by a single plasmid (preprint: Morgenthaler *et al*, 2019), to integrate the mutated donor onto the genome. The plasmid encodes *cas9* expressed under the constitutive Pro1 promoter (Davis *et al*, 2011) and the lambda Red recombination genes *exo*, *beta,* and *gam*

expressed using the heat-inducible pL promoter, induced by heat shock at 42°C (Yu *et al*, 2000). Similar to several lambda Red recombination systems (Sharan *et al*, 2009), the plasmid also has the temperature curable pSC101 origin of replication, which replicates at 30°C and is cured from the cells at 37°C (Phillips, 1999). In standard lambda Red recombineering protocols, the lambda Red recombination system is induced for 15 min prior to recombination (Sharan *et al*, 2009). Since the induction time is shorter than the replication time, the plasmid should be retained in most cells. The plasmid is cured prior to selection to remove the bulky plasmid that could impact cellular fitness and also to avoid fitness changes due to off-target effects of Cas9 (Hsu *et al*, 2013; Fig 1). We then sequenced the target region directly from the genome using deep sequencing before and after selection to quantify the frequency of mutations and estimate the distribution of fitness for the mutants in the library (Fig 1).

## There is a strong preference for integration of low-diversity sequences with PAM-proximal mutations

Cas9:gRNA-induced DSBs increase editing efficiency primarily by selecting for edited cells with the SPM introduced by recombineering (Cong *et al*, 2013; Jiang *et al*, 2013). Initially, we assumed that recombineering using a dsDNA substrate with single nucleotide changes may follow the same mechanism proposed for dsDNA-mediated gene replacement (Fig 2A). The lambda-Exo protein processes the dsDNA template into a single-stranded intermediate, which anneals to the Okazaki fragment using both ends by lambda-beta protein, and the gene replacement is completed by the native replication machinery (Mosberg *et al*, 2010; Fig 2A).

Beta can stably anneal DNA at both ends of the single-stranded repair intermediate with 1- to 2-kb-long non-homologous region using only 50 bp of flanking homology (Fig 2A; Maresca *et al*, 2010; Mosberg *et al*, 2010). Therefore, we expected that using a recombination template with single nucleotide changes, interactions between the annealed flanking homology may not have a significant impact on recombination, and the efficiency of recombination would be similar regardless of the number of mutations in the donor sequence. We used a 330-bp region in the *galK* gene as the target and developed two error-prone PCR donor libraries with high diversity and low diversity that contained 92 and 66% sequences with one or more mutations in addition to the synonymous PAM mutation (SPM), respectively (Fig 2B and Appendix Fig S1). The high-diversity donor contained a mean of 3–4 mutations per donor sequence, and the low-diversity donor contained a mean of 1–2 mutations per donor sequence (Appendix Fig S1). Hereon, we refer to the percentage of sequences with mutations in addition to the SPM as the mutation efficiency. As expected, after integration on the genome mutation efficiency with the high-diversity donor library was higher than the low-diversity donor (Fig EV1A). While in the high-diversity donor the sequences with 1–5 mutations were uniformly distributed, we observed a substantial bias toward sequences with 1 (only SPM) and 2 (1 mutation in addition to the SPM) on the genome (Fig 2C). Biased preference for sequences with fewer mutations was also observed with the low-diversity library (Fig EV1B). Contrary to our expectations, the efficiency of recombination decreased with an increasing number of mutations.

                                                                                      

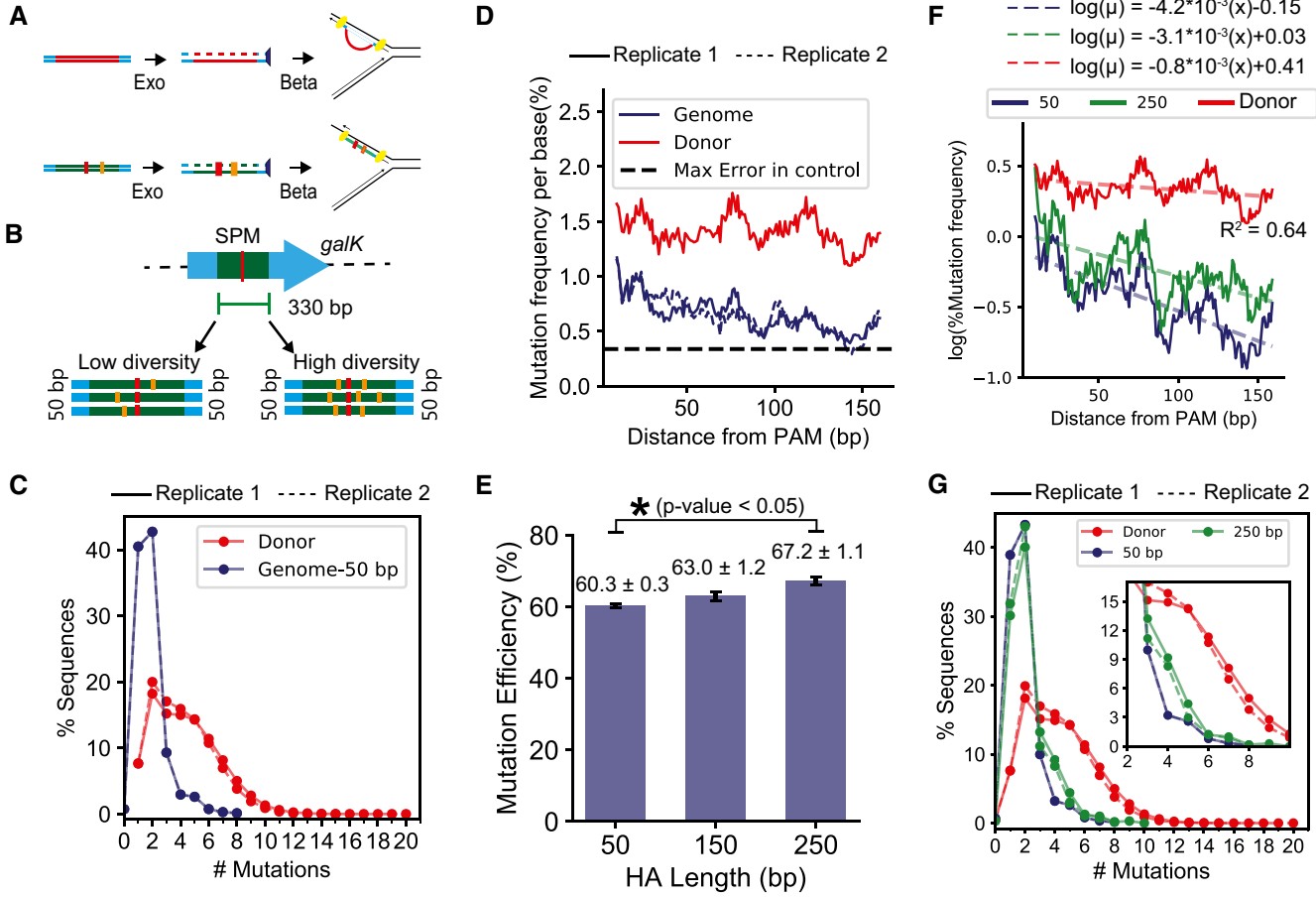

**Figure 2.** **Impact of donor diversity and homology arm length on mutation efficiency.**

A   Proposed mechanism for dsDNA-mediated gene replacement using recombineering occurs via a single-stranded intermediate (Mosberg et al, 2010) (top). We assume that lambda-mediated recombination of the CREPE substrate with limited mutations may also follow this proposed mechanism (bottom).

B   In our experiments to optimize CREPE, we target a 330-bp-long region within the galK gene. Initial tests were performed using high-diversity (mean 3–4 mutations per donor sequence) and low-diversity (mean 1–2 mutations per donor sequence) donors (Materials and Methods) with 50-bp-long unmutated end homologies.

C   A comparison of percentage sequence variants categorized by the number of mutations (x-axis) between the high-diversity donors before (red) and after (blue) integration on the genome. #Mutations = 1 corresponds to sequences with only the SPM. The experiments were performed in biological replicates. The trends for the replicates are represented by solid (—) and dashed (--) lines, respectively.

D   Change in mutation frequency per base (%) and percentage of sequences with a mutation at the position, using the high-diversity donor, are represented as rolling mean over 10 bases versus the distance from the PAM measured in base pairs. The experiments were performed in biological replicates. The trends for the replicates are represented by solid (—) and dashed (--) lines, respectively.

E   Comparison of % mutation efficiency determined by deep sequencing the genome after integrating the high-diversity donor using end homology of lengths of 50, 150, and 250 bp. Significant changes, determined as P-value < 0.05 for 1-tailed Student's t-test, are demonstrated using the *. Each value represents the mean, and error bars represent standard deviation for biological replicates.

F   Comparison of change in mutation frequency per base (%) and percentage of sequences with a mutation at each position, using the high-diversity donor, are represented as rolling mean over 10 bases versus the distance from the PAM (base pairs) for recombination of the high-diversity donor using 50- and 250-bp end homology. The dashed lines represent an exponential decay model fitted to quantify the decrease in mutation frequency with distance from PAM. The equations on top represent the fitted equation, and the R-squared value is the lower R-squared value of the 2 fits.

G   A comparison of percentage sequence variants categorized by the number of mutations (x-axis) for the high-diversity donor before (red) and after genome integration using 50-bp (green)- and 250-bp (blue)-long end homology. The inset highlights percent of sequences with 3–9 mutations in addition to the SPM. #Mutations = 1 corresponds to sequences with only the SPM. The experiments were performed in biological replicates. The trends for the replicates are represented by solid (—) and dashed (--) lines, respectively.

Additionally, if the end homology is sufficient for beta-mediated annealing (Fig 2A), the position of mutations within the target should not impact recombination efficiency (Li et al, 2013). While the mutation frequency in the donor was consistently high, we observed a decrease in the mutation frequency per residue with increasing distance from PAM on the genome (Figs 2D and EV1C).

Similar observations have been made when using double-stranded plasmid donors (Garst et al, 2017). With the high-diversity donor, the mutation frequency per residue exceeded the error frequency observed in unmutated regions across the entire target. However, with the low-diversity donor, high mutation frequencies per residue were observed predominantly within a 100-bp region around the

PAM (Fig EV1C). Mutations closer to the PAM had higher chances of being integrated on the genome as opposed to ones further from the PAM.

## Increasing end-homology length improves mutation efficiency for high-diversity donor

We posited that the PAM-proximal mutation bias was likely due to better annealing of sequences with mutations closer to the PAM because of longer uninterrupted end homology versus sequences with mutations distal to the PAM. Therefore, we evaluated whether the PAM-proximal bias could be alleviated by increasing end-homology length. With the high-diversity donor, we observed a $11.4 \pm 2.1\%$ increase in mutation efficiency by increasing the end homology from 50 to 250 bp (Fig 2E). For each HA length, the donor libraries were prepared by PCR amplification using the same error-prone PCR plasmid library, but with different primers to obtain different homology arm lengths. We used very high template concentrations, a high-fidelity polymerase, and low and same number of amplification cycles for each PCR. Therefore, the mutation efficiency of each donor library was expected to be consistent and similar to that of the plasmid error-prone PCR library. Therefore, we assumed that the variation in mutation efficiency on the genome due to variation in mutation efficiency of the donor was unlikely. To test whether the PAM-proximal mutational bias reduced with an increase in end-homology length, we quantified the decrease in mutation frequency with increasing distance from PAM as an exponential decay:

$$\mu = \mu o * \exp(-\lambda * x)$$

where $\mu$ is the mutation frequency, $\mu o$ is the maximum mutation frequency, $\lambda$ is the decay rate by position, and $x$ is the distance from PAM. The $\lambda$ after genome integration with 50-bp-long end homology with the high-diversity donor was significantly higher than that of the donor library itself. This demonstrated that the transformation led to a biased distribution of mutations as a function of distance from the PAM site. For the high-diversity donor library with an increase in end-homology length, there was a significant increase in $\mu o$ (ANOVA $P$-value of interaction = $10^{-16}$) and a slight but significant change in $\lambda$ as well (ANOVA $P$-value of interaction = 0.005) (Fig 2F). We observed a significant increase in the percentage of sequences with higher diversity (number of mutations in addition to the SPM > 2) on the genome (Fig 2G) ($P$-value for chi-squared test < $10^{-16}$). Increasing the HA length did not substantially reduce the PAM-proximal mutation bias but improved the mutation efficiency by improving recombination of donor sequences with a higher number of mutations per sequence on the genome. This was corroborated by the observation that for the low-diversity library, which lacked high-diversity sequences in the donor to begin with (Appendix Fig S1), the mutation efficiency and per-base mutation frequency did not change with the increase in the HA length (Fig EV1D and E).

## Inhibiting *mutL* improved mutation efficiency

Replication forks with beta-annealed ssDNA are usually resolved by native DNA polymerases and ligases (Sawitzke et al, 2011; Li et al,

2013; Fig 2A). Mismatches between the wild-type sequences and the recombination substrates are corrected by methyl-directed mismatch repair (MMR) to reduce recombination efficiency (Costantino & Court, 2003; Sawitzke et al, 2011). Therefore, deleting *mutL* or *mutS* genes improves recombineering efficiency with mutagenic single-stranded oligos (Costantino & Court, 2003; Sawitzke et al, 2011). However, the background mutation rates can increase significantly in *ΔmutS* and *ΔmutL* strains (Isaacs et al, 2011; Nyerges et al, 2014). Recently, the background mutation rate was significantly reduced by temporarily co-expressing a dominant-negative allele of MutL, MutL-E32K, with the lambda Red recombination proteins by heat shock at 42°C using the pL promoter (Nyerges et al, 2016). We cloned the *mutL-E32K* gene similarly in the plasmid encoding *cas9* and lambda Red recombination proteins (Fig 3A) and compared the mutation efficiency in the absence and presence of MutL-E32K using the high-diversity library with 250 bp HA. In the presence of MutL-E32K, the non-PAM editing efficiency improved by $24.2 \pm 0.8\%$, (Fig 3B) and we also observed an increase in mutation frequency per position across the target (Figs 3C and 2E). Expression of MutL-E32K significantly increased the maximum mutation frequency ($\mu o$, ANOVA $P$-value of interaction = $10^{-16}$ Fig 3C) and decreased the PAM-proximal positional bias of mutations (reduction in $\lambda$, ANOVA $P$-value of interaction = $10^{-16}$ Fig 3C).

Next, we evaluated whether the effects of increase in homology arm length and expression of MutL-E32K were mutually exclusive. If the improvement in editing through each occurred via independent mechanisms, a decrease in the HA length while conditionally expressing MutL-E32K would decrease editing efficiency. Therefore, we repeated editing with the Cas9+lambda Red recombination+MutL-E32K system using the high-diversity donor with 50-bp-long HA. Interestingly, we observed no significant difference between editing using 250-bp-long HA ($81.2 \pm 0.82\%$) and 50-bp-long HA ($79.9 \pm 0.82\%$; $P$-value = 0.1, Appendix Fig S2). This suggests that increase in HA length and expressing MutL-E32K are not mutually exclusive. Any limitations in recombination with 50-bp-long end-homology length can be overcome by only inhibiting MutL.

The specificity of MMR system varies with the various mismatch substrates (G-T, A-C, A-A, G-G > T-T, T-C, A-G >> C-C) (Lahue & Modrich, 1988; Modrich, 1991), which results in a significant bias in the recombination efficiency for different mutations using single-stranded oligos (Costantino & Court, 2003). These biases are also observed with Cas9:gRNA DSB-mediated recombineering in bacteria and can be reduced by inhibiting MMR (Li et al, 2015). Since inhibiting MMR significantly improved mutation efficiency using CREPE, we compared the frequencies for 12 possible base changes with and without MutL-E32K to evaluate whether such biases were eliminated. To our surprise, the frequency for different mutations was comparable in the presence and absence of MutL-E32K (Fig 3D). Moreover, the bias in base changes observed on the genome was comparable to the bias observed in the donor library (Fig 3D). This suggested that inhibiting MutL may follow an alternate mechanism to improve mutation efficiency.

The presence of MutL-E32K decreased the percentage of sequences with 1–2 mutations as well as increased the percentage of genomic sequences with more mutations per sequence (> 2 mutations in addition to SPM) (Fig 3E). Therefore, MutL-E32K reduced

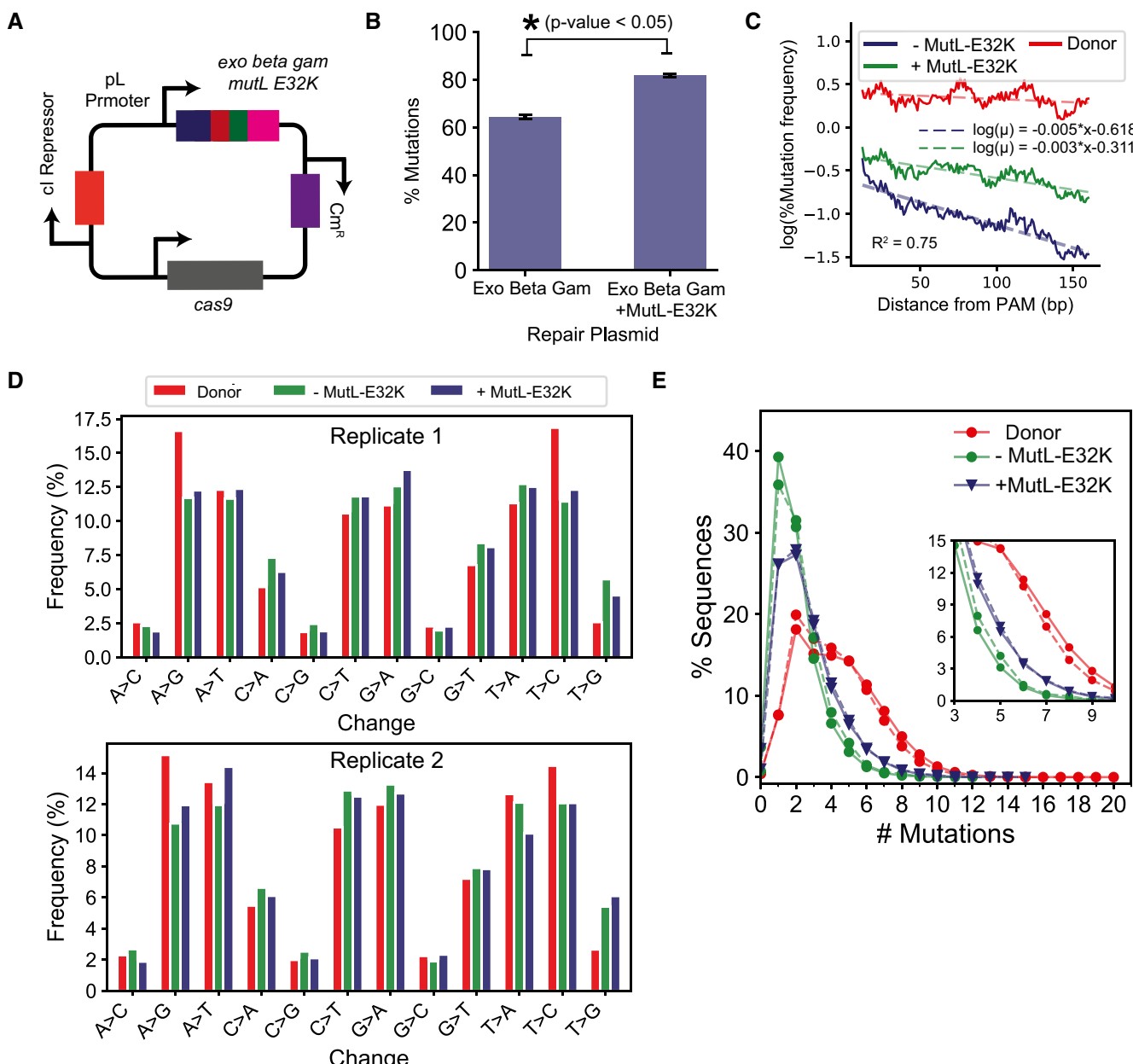

**Figure 3.  Impact of blocking methyl-directed mismatch repair (MMR) on mutation efficiency.**

A   Plasmid map where the *mutL-E32K* gene was placed under the temperature-inducible pL promoter in the lambda recombination operon as described by Nyerges *et al* (2016).

B   Comparison of % mutation efficiency determined by deep sequencing the genome after integrating high-diversity donor in the presence and absence of MutLE32K. Significant changes, determined as *P*-value < 0.05 for 1-tailed Student's *t*-test, are demonstrated using the *. Each value represents the mean, and error bars represent the standard deviation for biological replicates.

C   Comparison of change in mutation frequency per base (%) and percentage of sequences with a mutation at the position, using the high-diversity donor, are represented as rolling mean over 10 bases versus the distance from the PAM (base pairs) for recombination in the presence (blue) and absence (green) of MutLE32K. The dashed lines represent an exponential decay model fitted to quantify the decrease in mutation frequency with distance from PAM. The equations on top represent the fitted equation, and the *R*-squared value is the lower *R*-squared value of the 2 fits.

D   Comparison of percentage occurrence of individual base change combinations between the high-diversity donor (red) and the genome after recombination in the presence (blue) and absence (green) of MutLE32K. The bars represent mean percentage occurrence for individual base changes, and error bars represent the standard deviation for two biological replicates of next-generation sequencing data.

E   A comparison of percentage sequence variants categorized by the number of mutations (*x*-axis) between the high-diversity donor before (red) and after genome integration in the presence (blue) and absence (green) of MutLE32K. The inset highlights percent of sequences with 3–9 mutations in addition to the SPM. #Mutations = 1 corresponds to sequences with only the SPM. The experiments were performed in biological replicates. The trends for the replicates are represented by solid (—) and dashed (--) lines, respectively.

the bias toward incorporation of low-diversity sequences. This suggests that the presence of MutL-E32K may improve the recombination efficiency of templates with higher number of mutations.

## The CREPE technology allows efficient mutagenesis of regulatory and essential genes in *Escherichia coli*

We next measured the mutation efficiency using the optimized CREPE strategy for targeting genes that are essential and involved in global regulation. We targeted 230-bp-long regions in the genes: *crp*: a non-essential gene encoding a global metabolism regulator that controls the expression of hundreds of proteins in *E. coli* (Görke & Stülke, 2008), *rpoB*: an essential gene that encodes the beta subunit of RNA polymerase, and *mreB*: an essential gene that encodes a cytoskeletal protein (Fig 4A). We targeted a shorter window for these essential genes compared to *galK* because with the shorter window size, the entire targeted region can be covered by a single read using a paired-end Miseq. The complete overlap between the forward and reverse reads allows for improved read quality and reduces error. For each of the target genes, we observed a high mutation frequency per base across the targeted region (Fig 4A). However, the mutation efficiency for these targets was significantly lower than *galK* (Fig 4A). We observed a higher percentage of variants with only the synonymous PAM mutation in the donor libraries for these targets, which could explain the lower mutation efficiency in these targets compared with *galK* (Fig 4A). We used the same error-prone PCR protocol for each of these targets as *galK* (Fig 4B). Since the mutation frequency using error-prone PCR is directly proportional to amplicon length, the mutation efficiency was lower in the donors for the new targets compared with *galK*.

However, between these targets, the mutation efficiency for *rpoB* and *mreB* was significantly lower than that of *crp*, even though they had the same target size. The lower editing could be due to the essentiality of *rpoB* and *mreB*. Therefore, deleterious mutations would not be tolerated on the genome. We expected that sequences with higher number of mutations would deplete for essential genes because a higher number of mutations are more likely to be deleterious. Therefore, we compared the distribution of variants based on number of mutations per sequence (Fig 4B). While the distribution was comparable in the donor, we observed fewer sequences with greater than two mutations in addition to the PAM in *rpoB* and *mreB* compared with *crp* (Fig 4B). To further validate our hypothesis, we also compared the percentage type of mutation (synonymous, non-synonymous, and stop codons) between the donor and the genome. For *rpoB* and *mreB*, the percentage of synonymous substitutions increased, the percentage of non-synonymous substitutions decreased, and substitutions leading to stop codons diminished (Fig 4C). This observation is expected as non-synonymous mutations are more likely to have a functional impact than synonymous mutations and stop codons in essential genes would be deleterious. Few sequences with stop codons occurred in *rpoB* and *mreB* likely due to sequencing and PCR errors (Fig 4C). Therefore, although we obtained high mutation efficiency for essential genes, the essentiality of these genes impacted the spectrum and distribution of mutations.

For successful deep mutational scanning, multiple substitutions should be possible at each residue within the target for adequate sampling of the sequence space. In addition to adequate number of

substitutions, there should be an adequate number of counts associated with each substitution for efficient accurate fitness estimates. Therefore, we next analyzed the average codon substitutions and position-wise variant frequencies for each of the targets. For each gene library, we scraped ~ 20,000 colonies. We observed several substitutions at each targeted residue within each target gene (Fig 4D), and we observed 7.75, 7.5, and 7.55 mean substitutions per codon for *crp*, *mreB*, and *rpoB*, respectively (Fig 4D). The variant counts for substitutions at each position primarily varied between $10^2$ and $10^4$ counts per variant (from ~ $2*10^6$ total counts) with 439, 613, and 692 median reads per variant for *crp*, *mreB*, and *rpoB*, respectively (Fig 4D). Therefore, using CREPE we were able to successfully develop mutation libraries over the entire targeted sequence space with significant counts associated with variants for fitness mapping, of several genes in *E. coli*.

## CREPE can be used to identify potentially deleterious mutations in essential genes

Since we observed a significant depletion of stop codons (loss-of-function mutations) for essential genes (Fig 4C), we posited that we could compare the change in frequency of variants between the donor library and after integration on the genome to identify deleterious mutations for essential genes in *E. coli* (Fig EV2A). We measured fitness of substitution as log change in frequency between the donor library and the frequency after integration on the genome (Fig EV2B). We compared the distribution of fitness after integration for non-synonymous, synonymous, and stop codon substitutions for the non-essential *galK* gene and the essential *rpoB* gene (Fig EV2A). In *galK*, we observed that the distribution of fitness effects for non-synonymous, stop codons, and synonymous substitutions overlapped with each other (Fig EV2B). This was expected as the nature of substitution in a non-essential gene should not impact cell survival after integration and consequently fitness. However, for *rpoB*, in stark contrast, the distributions for synonymous and stop mutations did not overlap (Fig EV2B). The distribution of fitness for synonymous mutations was centered slightly below 0 and that for the stop codons centered around −4 (Fig EV2B). Therefore, in the case of essential genes we observed a clear signal to differentiate between deleterious and non-deleterious mutations in the gene (Fig EV2B). The distribution of fitness of non-synonymous mutations in *rpoB* had a peak overlapping with the synonymous mutations and a long tail with several mutations with high negative fitness that was comparable to the fitness of the stop codons (Fig EV2B). We used the distribution of fitness for stop codons as a reference to find 25 substitutions that could be potentially deleterious in *rpoB* (Appendix Table S1). Therefore, we can use the CREPE technology to identify potentially deleterious substitutions in essential genes in *E. coli*. Identification of such substitutions could be important for identification of functional residues in poorly characterized essential genes.

## Fitness estimates of mutations that confer resistance against rifampicin

In order to demonstrate high-throughput fitness scoring, we decided to study mutations in *rpoB* that confer resistance against rifampicin. Although each of our target regulatory and essential genes have

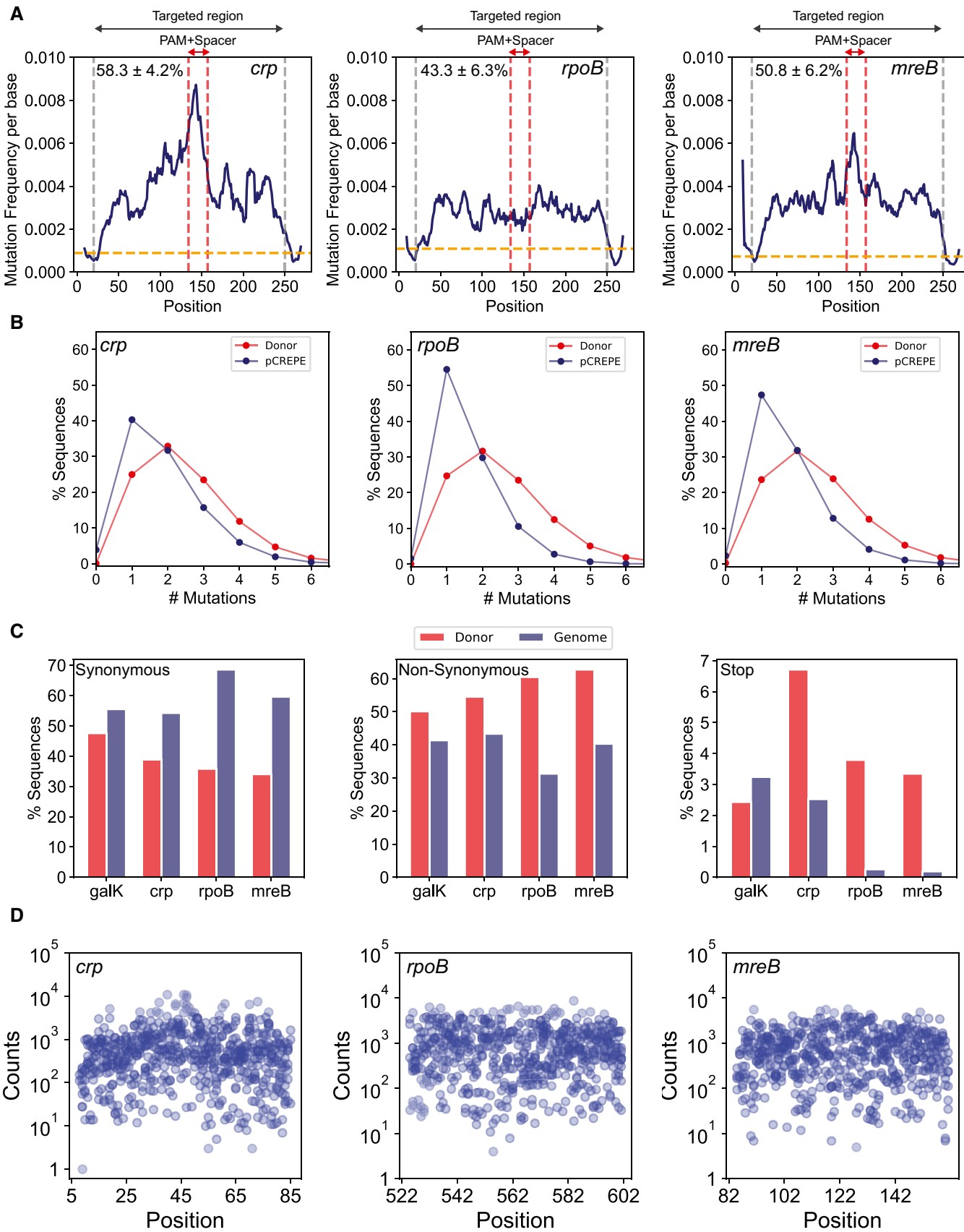

**Figure 4.**

**Figure 4.  CREPE-mediated targeting of different genomic loci.**

A  The change in mutation frequency per base and fraction of sequences with a mutation at the position, using the high-diversity donor, are represented as rolling mean over 10 bases along the length of the targeted region (within gray vertical lines, excluding the SPM) for each targeted locus *crp* (left), *rpoB* (middle), and *mreB* (right). The red vertical dashed lines highlight the PAM+spacer region within the target. The percentages on the top represent mutation efficiency for different targets.

B  A comparison of percentage sequence variants categorized by the number of mutations (*x*-axis) between the donor (red) and after integration on the genome (blue) for genes *crp* (left), *rpoB* (center), and *mreB* (right).

C  A percentage of sequences with synonymous mutations (left), non-synonymous mutation (center), and stop codon mutations (right) between the donor (red) and after integration on the genome (blue) for the four target genes.

D  The $\log_{10}$(counts) (*y*-axis) associated with a variant (variant number *x*-axis) from the *rpoB* library among variants with only one mutation in addition to SPM per position for *crp* (left), *rpoB* (center), and *mreB* (right).

been found to play an important role in evolution, we chose to focus the validation of fitness estimates on *rpoB* for two reasons. Firstly, resistance to antibiotics has a clean survival/non-survival outcome that is easily interpretable. Secondly, resistance to rifampicin has been extensively characterized over the past three decades which allowed adequate interpretation of our fitness estimates. Rifampicin is an essential drug used for chemotherapy against tuberculosis (Conde & Lapa E Silva, 2011), which is currently the most prevalent and fatal infectious disease (Global Tuberculosis Report—World Health Organization). Rifampicin acts by binding to the catalytic core of RpoB to inhibit the elongation step in transcription. Clinical resistance to rifampicin arises due to mutations predominantly in three regions of the *rpoB* gene termed as rifampicin resistance-determining regions, RRDRs I (residues 507–533), II (residues 563–572), and III (residue 687) (Fig 5A) (Sandgren *et al*, 2009). We decided to target a 90-amino acid region in RpoB which included half of RRDR I and the entire RRDR II, covering residues, which account for ~ 90% of the known resistance-causing mutations (Fig 5A) (Sandgren *et al*, 2009). Since the resistance mechanisms within this region have been extensively characterized, we correlated the fitness estimates for mutations using CREPE to the previous biochemical and epidemiological findings. We designed the recombination template such that RRDR II was PAM-proximal and RRDR I was PAM-distal to validate fitness estimates for mutations along the entire target length (Fig 5A).

As we discuss later, the concentration of rifampicin can impact the fitness of mutations. Therefore, we compared the change in frequency of mutants in the library immediately after construction ($t_0$) as well as after growth on three different concentrations of rifampicin (10 μg/ml: which is slightly lower than the MIC of 12, 100 μg/ml: the standard concentration for selection in the laboratory, and 50 μg/ml: an intermediate between the two) as well as in the absence of rifampicin in biological triplicates ($t_f$) (Fig 5B). At each of these concentrations, the SPM used in the template for the library did not alter fitness, measured as colony-forming units (CFUs) per ml of cell culture, as compared to wild-type cells (Fig 5C). The number of resistant CFUs was two orders of magnitude higher for the library at 10 μg/ml and four orders of magnitude higher at 50 and 100 μg/ml than the wild-type cells and cells with just the SPM (Fig 5C). A significant increase in mutation frequencies was observed within both RRDRs demonstrating that the library covered mutations along the entire target (Fig 5D). In comparison, such significant peaks in mutation frequency were not observed in the no-drug control, where the library was grown overnight (for the same number of doublings as the rifampicin plates) on LB plates without any rifampicin (Fig 5D). Interestingly, we also observed an increase in the number of resistant CFUs when we plated cells in

which the gRNA was co-transformed with a template having only the SPM on different concentrations of rifampicin (Fig 5C). While over 100 single- and double-resistant mutants were identified with the *rpoB* library, only 4 unique mutations were identified with the SPM only template primarily located in the PAM-proximal RRDR (Appendix Fig S3). The resistance could be an outcome of unintended mutations introduced in the template DNA during PCR, or by error-prone polymerases expressed during the SOS response to DSBs at the target (Mallik *et al*, 2015). Regardless, we observed over an order of magnitude more resistant CFUs representing over 25-fold more unique variants in our error-prone library compared to the cells transformed with the SPM only template.

We next calculated the fitness of each mutant as the log-fold change in the frequency between $t_0$ and $t_f$ normalized to the change in frequency of a wild-type control to identify the mutations that conferred resistance (Fig 5E and Materials and Methods). The fitness values for the mutations common across all replicates correlated strongly (Appendix Fig S4) (Pearson's correlation coefficient between 0.81 and 0.98). We averaged the fitness scores for common mutations across replicates using the Fisher scoring iteration-based maximum-likelihood estimates (Materials and Methods; Rubin *et al*, 2017). We observed a bimodal distribution of fitness effects for all mutations at each rifampicin concentration (Fig 5E). The lower mode of the distribution of fitness aligned with the distribution of fitness for synonymous mutations in the population (Fig 5E). Since resistance to rifampicin is primarily caused by mutations that inhibit the interaction of rifampicin to the binding pocket, synonymous mutations are unlikely to confer resistance. Therefore, we used the synonymous mutations population as a control to differentiate between the sensitive and resistant mutations (Fig 5E). In comparison, we observed that the peak for distribution of fitness effects for most mutations was centered around 0 in the absence of rifampicin (Appendix Fig S5). We defined resistant mutants as mutations with fitness greater than three standard deviations of the mean fitness of synonymous mutations (Fig 5E).

## Beneficial mutations corroborate biochemical properties and epidemiological findings

We first studied the fitness effects of single non-synonymous mutations in the population. We estimated a fitness score for 355 single non-synonymous mutations across replicates covering 88 of the 90 targeted amino acid residues. We found 16 resistant mutations each for 100 and 50 μg/ml, and 37 mutations resistant at 10 μg/ml rifampicin (Fig 6A and Appendix Table S2). Of the 39 unique resistant mutations identified at different concentrations, 42% were already known to confer resistance and 81% of the mutations occurred in

sites known to confer resistance, predominantly at or adjacent to the rifampicin-binding pocket (Fig 6B and Appendix Table S2; Campbell *et al*, 2001; Sandgren *et al*, 2009; Zhou *et al*, 2013). We identified 22 new rifampicin resistance mutations (Fig 6B). We reconstructed nine of the new mutations and found that eight out of nine reconstructed mutants had a higher minimum inhibitory

concentration (MIC) than wild-type *E. coli* (Appendix Figs S6 and S7). Among the 88 targeted residues, positions H526, R529, S531, L533, and I572, each of which can either H-bond or form van der Waals interactions with rifampicin (Campbell *et al*, 2001), had significantly higher fitness as compared to other residues within the pocket (*P*-value < 0.0001 Student's *t*-test for 10, 50, and 100 µg/ml,

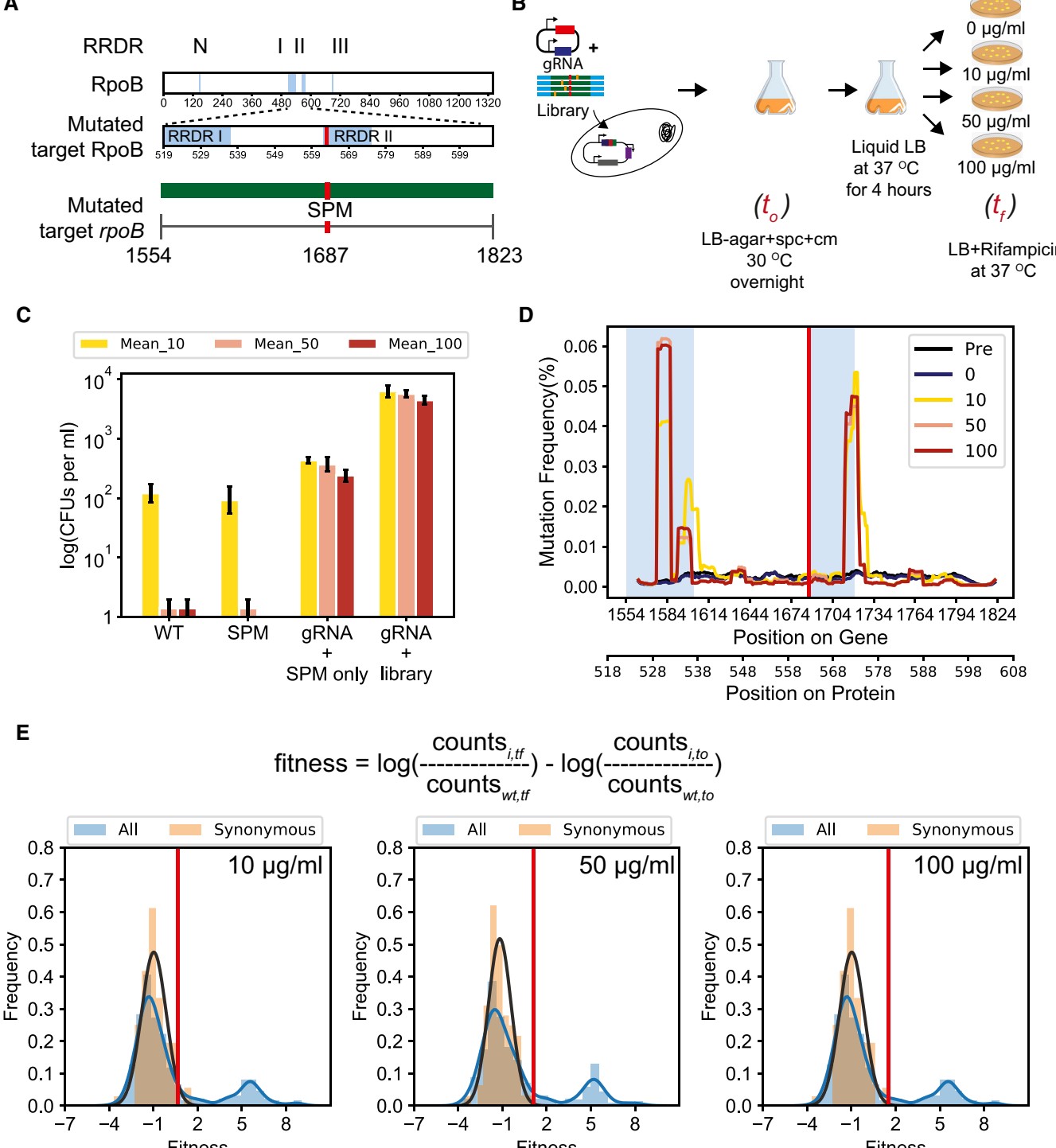

**Figure 5.**

**Figure 5.  CREPE-mediated mutagenesis of *rpoB* for resistance to rifampicin.**

A   Four distinct regions within *rpoB* (N, I, II, and III) are rifampicin resistance-determining regions (RRDRs). We used CREPE to make a library covering a 270-bp-long region in *rpoB* covering half of RRDR I and entire RRDR II.

B   Experimental setup for studying resistance to 3 different concentrations of rifampicin (10 μg/ml (yellow), 50 μg/ml (pink), and 100 μg/ml (red)) (Materials and Methods).

C   (Left to right) Cell growth determined as log(number of colony-forming units or CFUs per ml of culture) after growth on LB agar with different concentrations of rifampicin, 10 μg/ml (yellow), 50 μg/ml (pink), and 100 μg/ml (red), for wild-type *Escherichia coli* MG1655 (WT), a single colony of *E. coli* MG1655 + synonymous PAM mutation (SPM), a library of colonies recovered by scraping colonies developed after transforming cells with the *rpoB* targeting gRNA and only the SPM mutation (gRNA + SPM only), and with the *rpoB* targeting gRNA and only the *rpoB* error-prone PCR library (gRNA + library) (see Materials and Methods for details). The bars represent mean percentage occurrence for individual base changes, and error bars represent the standard deviation for three biological replicates.

D   The change in mutation frequency per base, fraction of sequences with a mutation at the position, represented as rolling mean over 10 bases along the length of the targeted region for the cells with the *rpoB* genomic error-prone PCR library at different concentrations of rifampicin, 10 μg/ml (yellow), 50 μg/ml (pink), and 100 μg/ml (red). The blue windows cover RRDR I and RRDR II in the target region as described in Fig 5A.

E   Distribution of fitness estimates for all mutations in the *rpoB* library (blue with blue line) and only synonymous mutations (salmon with the black line representing a fit for the normal distribution to estimate the mean and standard deviation of the distribution of fitness effects) in the *rpoB* library at different concentrations of rifampicin, 10 μg/ml (left), 50 μg/ml (center), and 100 μg/ml (right). The red vertical line indicates the threshold for fitness ≥ 2.96 times the standard deviations from the mean fitness of synonymous mutations.

Figs 6C, and EV3A and B). Substitutions at positions S531 and H526 that represent the majority of known mutations in clinical isolates (~ 41 and ~ 36%, respectively) had the highest mean fitness in the presence of rifampicin compared to all other positions (Fig EV3C; Sandgren *et al*, 2009). Therefore, the fitness estimates agree with known biochemical properties and clinical observations within the target.

**Fitness estimates help understand the biochemical basis for impact of rifampicin concentration on selection**

The evolution of resistance depends on the exposure profile of the microbe to the antibiotic (Gall *et al*, 2008; Palmer & Kishony, 2013; Wistrand-Yuen *et al*, 2018). We observed that more mutations were selected at subinhibitory concentrations (10 μg/ml) of rifampicin (Fig 6A and B, and Appendix Table S1) than at higher antibiotic concentrations (50 and 100 μg/ml). This is expected as there are more paths to resistance at subinhibitory antibiotic concentrations (Lindsey *et al*, 2013; Wistrand-Yuen *et al*, 2018). Interestingly, while all mutations resistant to rifampicin at higher antibiotic concentrations (50 and 100 μg/ml) were resistant at subinhibitory concentrations (10 μg/ml), a second population was observed that was resistant only to 10 μg/ml of rifampicin (Figs 6D and EV4A). Variants identified at 10 μg/ml of rifampicin had lower MIC values than variants identified at 100 μg/ml of rifampicin (Appendix Figs S6 and S7).

Rifampicin resistance occurs primarily by blocking the access of rifampicin to the binding pocket (Campbell *et al*, 2001). While all high-resistance-conferring substitutions were in residues directly in contact with rifampicin, one subset of substitutions with resistance to lower rifampicin concentrations occurred in residues next to the binding pocket (Fig 6B). These residues likely impact rifampicin binding by perturbing the structure of the binding pocket but would have weaker effect compared with the substitutions in the binding pocket. Substitutions of the H-binding residue S531 with residues with short side chains such as S531C and S531A conferred low resistance, whereas bulkier substitutions such S531L and S531N conferred resistance to higher concentrations of rifampicin (Fig EV4B). Similarly, for van der Waals forming residue I572, less-bulky substitutions such as I572S and I572T conferred resistance only to low rifampicin concentration as compared to the bulkier

high-resistant changes I572L, I572F, I572Y, and I572H (Fig EV4C). While all the substitutions would prevent H-bond or the van der Waals interaction, resistance increased with bulkiness of the side chain likely because of increased steric interactions. The bulkier substitutions in residue S531 have been shown to push the fork loop adjacent to it, which increases the solvent exposure of rifampicin-binding pocket and therefore drastically increases resistance to rifampicin (Molodtsov *et al*, 2017). Therefore, the less-bulky residues, which are less likely to perturb the structure of the binding site, would be expected to have lower resistance. This demonstrates that resistance mechanisms for mutation at certain residues such as S531 and I572 are more complex than just steric inhibition. Our results corroborate a recent finding that high resistance is an outcome of several changes within the binding pocket (Molodtsov *et al*, 2017). Overall, the fitness estimates at different concentrations of rifampicin sensitively capture the biochemical contribution of different substitutions based on their position and type of change. Similar sensitive fitness estimates at different drug concentrations may improve our biochemical understanding of resistance.

At times, variants with lower MICs are preferentially selected at lower rifampicin concentrations in the laboratory and clinic (van Ingen *et al*, 2011; Lindsey *et al*, 2013; Berrada *et al*, 2016). Mutations with high rifampicin resistance confer growth defect because they occur close to the catalytic site of RNA polymerase (Campbell *et al*, 2001) and consequently are detrimental to the cell (Brandis & Hughes, 2018). The prevalence of some lower MIC variants could be due to a trade-off between resistance and detrimental effects of mutations. Interestingly, at the rifampicin concentration of 10 μg/ml the maximum fitness for some mutations resistant to only 10 μg/ml was comparable to the maximum fitness for mutations selected at 50 and 100 μg/ml (Figs 6D and EV4A). These variants selected at lower rifampicin concentration had weaker resistance and lower MICs compared with the ones selected at 50 and 100 μg/ml (Appendix Figs S6 and S7). We posited that the comparable fitness could be due to the above-mentioned trade-off between resistance and growth defects. Since the substitutions in weakly resistant mutations with a low MIC are less-bulky and/or away from the active site (Fig EV4B and C), they may have less detrimental effects. Therefore, at lower concentrations of rifampicin, the strongly resistant mutations have high fitness mainly due to their strong inhibition of rifampicin binding. However, at the same lower rifampicin

concentration the weakly resistant mutations may have their fitness equivalent to the strongly resistant mutations because of a relative growth advantage, despite the weaker inhibition. This hypothesis was further confirmed when we found that the low-resistance mutations grew significantly better than the high-resistance mutations in the absence of rifampicin (Student's *t*-test *P*-value < 0.01, Fig 6E). These observations explain why low-resistance mutations are preferred at lower concentrations of rifampicin, an observation made both in *E. coli* (Lindsey *et al*, 2013) and in *Mycobacterium tuberculosis* (Billington *et al*, 1999; Sander *et al*, 2002).

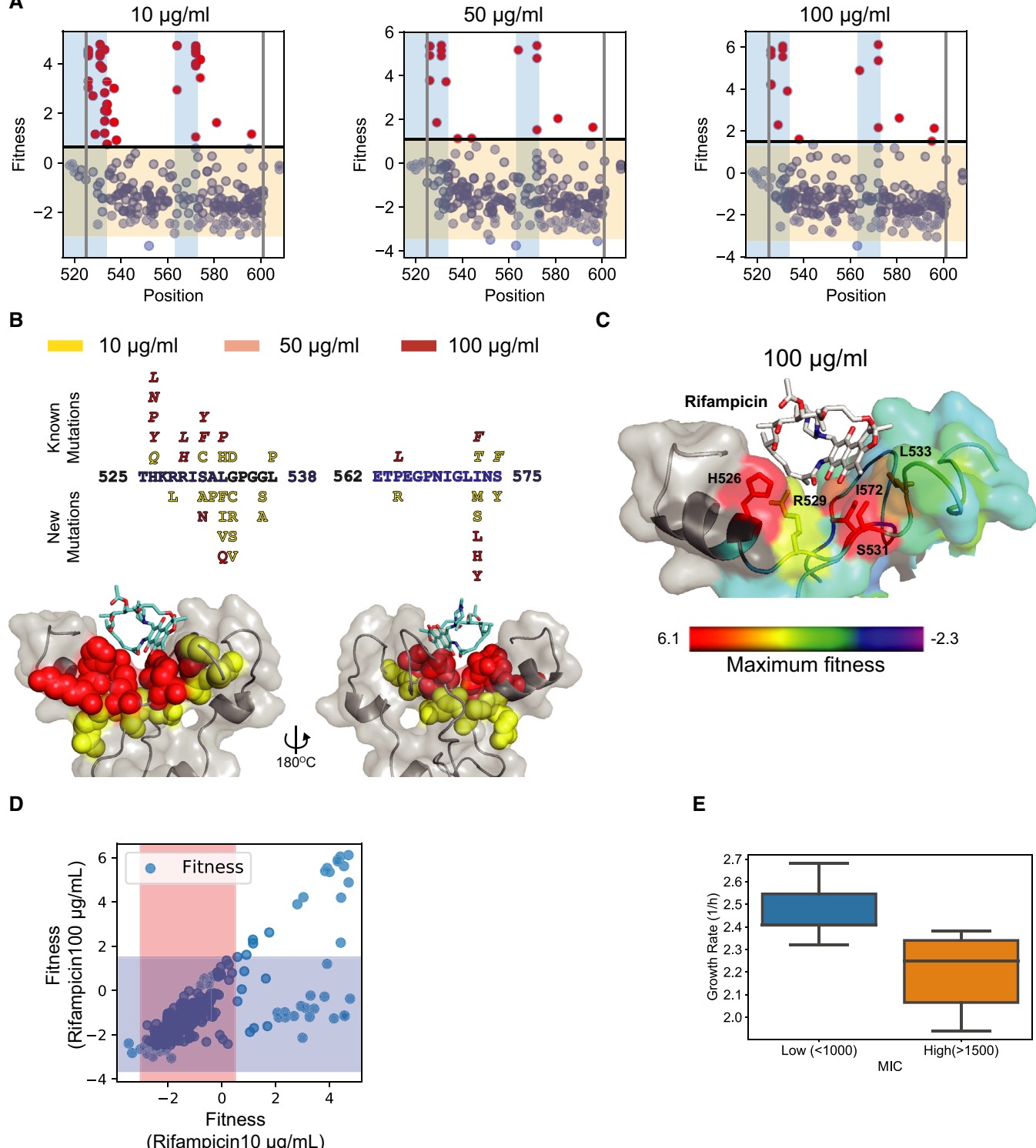

Figure 6.

**Figure 6. Position, rifampicin concentration, and structure-dependent trends in rifampicin resistance.**

A  Fitness for mutations (blue) at each position in the RpoB target at 10 μg/ml (left), 50 μg/ml (center), and 100 μg/ml (right). The gray lines represent the region within which mutagenesis was performed. The blue windows represent the RRDRs. The orange window represents the range of fitness values within 2.96 fitness standard deviations, plus and minus the mean fitness of synonymous mutations in the library. Any mutation with fitness greater than or equal to the mean fitness of synonymous mutations + 2.96*standard deviation of fitness of synonymous mutations was selected as resistant and represented using red dots.

B  Mutations within RRDR I (525–538) and RRDR II (562–575) identified at different rifampicin concentrations 10 μg/ml (yellow), 50 μg/ml (pink), and 100 μg/ml (red), with the known mutations above the target sequence and new mutations below the target sequence. The structure (cartoon + gray surface) represents rifampicin (sticks)-binding pocket mutations with spheres representing the mutations identified at 10 μg/ml (yellow), 50 μg/ml (pink), and 100 μg/ml (red).

C  The structure (cartoon + surface) represents rifampicin (white sticks)-binding pocket. The residues are colored using a heat map (range below the structure) demonstrating the maximum fitness at each residue. Residues with sticks form H bonds or van der Waals interactions with rifampicin (Campbell *et al*, 2001). The fitness scores were evaluated at 100 μg/ml rifampicin. The structure with PDB code 5UAC was used for representation (Molodtsov *et al*, 2017).

D  Correlation of fitness estimated for the same mutant at 10 μg/ml rifampicin (x-axis) and 100 μg/ml rifampicin (y-axis). The orange and blue regions represent the window within 2.96 fitness standard deviations (on each side) around the mean fitness of synonymous mutations at 10 and 100 μg/ml, respectively.

E  A comparison of the distribution of growth rates for 8 variants with a lower minimum inhibitory concentration (MIC) (blue) and 8 variants with a higher minimum inhibitory concentration (MIC) (orange) to rifampicin upon growth in LB without rifampicin. Within the box, the central line represents the median, the bottom and top edges of the box represent the 25 and 75% quartiles, the bottom and top whiskers represent 9 and 91% percentiles, and the dots represent outliers.

Low-concentration antibiotic treatments are clinically relevant for long-term TB treatment using multiple antibiotics (van Ingen *et al*, 2011). The fitness estimates highlight that lower antibiotic concentrations allow selection of variants that have better growth in the absence of antibiotic than the growth of variants selected at high antibiotic concentrations. This explains the successful transmission of mutations selected in low-concentration treatments observed previously (van Ingen *et al*, 2011).

**Epistasis in double mutants**

DMS using CREPE not only allows the study of individual mutations, but also the analysis of combinations of mutations. Using the double mutants introduced using CREPE, we studied how the epistatic effects of single-point mutations affected the fitness of double mutants in *rpoB* in the presence of rifampicin. We plotted a correlation between the sum of fitness for individual mutations in the double mutant versus the actual fitness of the double mutant. This was done because the fitness formula we use is a measure of log fitness. The fitness correlation between the actual and predicted values would lie on a 45° diagonal in the case of no epistasis, below the diagonal in the case of negative epistasis (fitness of double mutant is lower than the sum of the individuals), and above the diagonal in the case of positive epistasis (fitness of double mutant is higher than the sum of the individuals). We split the mutations into three groups depending upon whether the individual mutations were either resistant or sensitive: both resistant, one resistant, and one sensitive and both sensitive (Fig 7A). Each group demonstrated a unique epistatic behavior (Figs 7A and EV5A). Epistasis occurs when the fitness of mutation combinations deviates from 0. However, there are errors associated with fitness estimates using deep sequencing (Materials and Methods). Therefore, we assigned epistasis signs to the mutation combinations when whole fitness deviated from the sum of mutations after accounting for the error in the fitness estimates for individual mutations (Materials and Methods).

We observed rare occurrences of reciprocal sign epistasis, where individual mutations were not resistant to a particular concentration of rifampicin, but resistance emerged in the combination. For example, in most of the combinations with resistance to 100 μg/ml of rifampicin, such as G534S/S574F, S574Y/L533I, and G537C/I572M,

individual mutations had high fitness at 10 μg/ml of rifampicin and not at higher rifampicin concentrations. Individually, reconstructed G534S/S574F and S574Y/L533I had MIC values of > 1,500 and 500 μg/ml, respectively (Appendix Fig S8). In the case of I572N/S574Y, the mutations occurred at positions known to confer rifampicin resistance (Fig 7B). The mutations were bulky substitutions in residues close to each other near the rifampicin-binding pocket. Therefore, the combination of these mutations is likely to synergistically change the structure of the binding pocket and the reciprocal sign epistasis is observed. In each case, the reciprocal sign epistasis was justified. Similar to our observations with different rifampicin concentrations (Fig 6) and previous studies (Molodtsov *et al*, 2017), these findings highlight that resistance to rifampicin may be an outcome of several complex changes within the binding pocket and not just steric inhibition.

Over 70% of the double mutations with combinations of individual beneficial mutations had negative (antagonistic/diminishing returns epistasis) epistasis, where the actual fitness was lower than the fitness sum. Antagonistic epistasis between beneficial mutations is common across different systems (Khan *et al*, 2011; Tokuriki *et al*, 2012). However, higher fitness of the double mutants as compared to each individual mutation was unexpected in several cases. Single mutations such as H526L, P564L, and I572F individually have a MIC > 1,500 μg/ml (Appendix Figs S6 and S7), 15-fold higher than the selection concentration of 100 μg/ml. Moreover, these beneficial mutations were individually detrimental to the host in the absence of rifampicin (Fig 6E). The observed improvement in fitness of these double mutants likely occurred because the beneficial mutations compensated for each other's fitness costs due to detrimental effects. This observation has been made previously for individual fitness estimates for double mutants in *Pseudomonas aeruginosa* (Hall & MacLean, 2011). Additionally, compensatory mutations in RpoB, that confer resistance to rifampicin, have also emerged in evolved *M. tuberculosis, E. coli,* and *Salmonella enterica* (Reynolds, 2000; Brandis *et al*, 2012).

The compensatory fitness improvement may also explain the high positive epistasis (in 78% of the combinations) between sensitive and resistant mutation combinations. Due to the fitness cost of resistant mutations, compensatory mutations are known to drive transmission of drug-resistant bacteria across populations (Comas *et al*, 2011; Zhang *et al*, 2013; Casali *et al*, 2014). Our data reveal

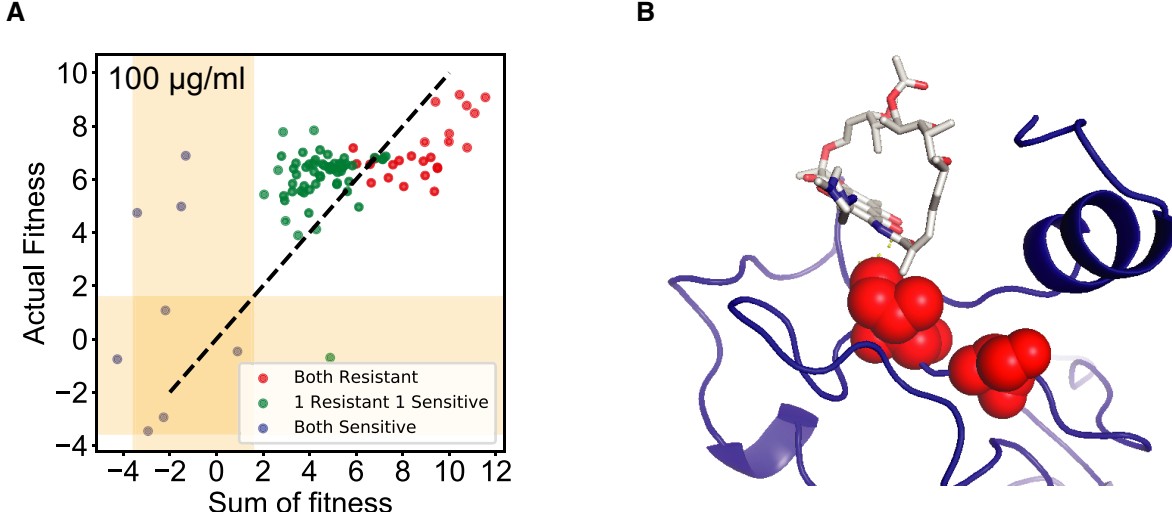

**Figure 7. Trends for epistasis observed in double mutants.**

A Comparison of the actual fitness determined by CREPE and the predicted fitness determined as the sum of the fitness for individual mutations in the double mutants for resistance to 100 μg/ml of rifampicin. The double mutants were categorized as combination of 2 resistant mutations (red), 1 resistant and 1 sensitive mutant (green), and 2 sensitive mutations (blue).

B Mutant combination at residues I572 and S574, that are individually non-resistant to rifampicin at all concentrations, demonstrates reciprocal sign epistasis in combination. The structure with PDB code 5UAC was used for representation (Molodtsov et al, 2017).

there may be a high frequency of available compensatory mutations within a 90-bp region of the RpoB. This finding corroborates previous observations in laboratory evolution where compensation more often than not has occurred through mutations within the *rpoB* (Reynolds, 2000; Comas *et al*, 2011; Brandis *et al*, 2012; Casali *et al*, 2012). Here, by revealing the breadth of the target for compensation within *rpoB*, we suggest that prolonged usage of rifampicin may result in the emergence of RpoB alleles with high resistance and compensated fitness cost, which could dramatically enhance the propagation of resistance.

## Discussion

In summary, we describe CREPE for introducing mutations using an error-prone PCR donor containing the common SPM and a single gRNA to introduce targeted diversity on the genome (Fig 1). Recombineering using a linear dsDNA donor with single nucleotide changes is poorly understood. We found that recombination rate decreased with increasing sequence diversity despite having long homology arms. By modifying donor mutation rate, homology arm length, and mismatch repair, we significantly increased the library editing efficiency and reduced library biasing (Figs 2 and 3). Our results also suggest that recombination may follow an alternate RecA-mediated template switch model (Fig 3). Using CREPE, we were able to introduce a library of mutations across different essential and non-essential genomic loci (Fig 4). The high-efficiency and diverse mutant library allowed us to perform DMS for naturally accessible mutations that confer resistance to rifampicin (Figs 5–7).

We initially assumed that lambda Red-mediated recombination using a template with limited mutations may follow a single-strand

intermediate-dependent model proposed for dsDNA-mediated gene replacement (Fig 2A). Using the model, we hypothesized using a 50 bp of end homology the recombination efficiency should be similar between templates irrespective of number and position of mutations within the sequence (Fig 1). However, contrary to our hypotheses based on the previous model, we observed that the recombination depended on PAM proximity and the number of mutations in the template (Fig 2). According to the previous model, DNA polymerases are used to replicate using the mutated recombination donor as a template. Consequently, mutation-specific biases (G-T, A-C, A-A, G-G > T-T, T-C, A-G >> C-C) occur due to repair by the methyl-directed mismatch repair machinery after recombination (Lahue & Modrich, 1988; Modrich, 1991). However, using a dsDNA with point mutations as recombination template we did not observe such mutation biases on the genome (Fig 3). We think that an alternate template-switching model proposed in a recent review by Murphy (Murphy, 2016) may explain these observations for recombineering using a template with limited mutations. According to this model, as a final transfer step the native RecA recombination machinery of *E. coli* recombines a broken chromosome with the mutations of the donor template to an intact chromosome (Murphy, 2016). RecA-mediated recombination can form crossover products between homologous sequences within as few as 8-bp-long sequences (Hsieh *et al*, 1992). The cells only need synonymous PAM mutations to prevent Cas9:gRNA-induced cell death. So, as the distance between the SPM and targeted mutation increases, the chances of decoupling of the target mutations and the SPM increase to introduce a PAM-proximal bias (Fig 2). According to this mechanism, the replication machinery uses the recombination donor as a template for replication due to which the mutations in the donor are not identified as mismatches (Murphy, 2016). This could explain why the

mutation bias due to MMR is not observed. MutS and MutL block branch migration during RecA recombination (Worth *et al*, 1994; Tham *et al*, 2013). The inhibition of RecA recombination by MutL is stronger for recombination templates with a higher number of mismatches (Tham *et al*, 2013). This could explain the drop in recombination efficiency with an increase in the number of mutations per sequences (Fig 2). Additionally, this may also explain the increase in recombination efficiency with more diverse sequences when we express MutL-E32K to inhibit MutL (Fig 3). Lastly, RecA-mediated recombination is more sensitive to HA length over 50 bp than lambda Red-mediated recombination, which explains the improvement in recombination with increase in HA length over 50 bp (Fig 2). Further investigation of this model is needed to identify additional targets and strategies to improve CREPE.

Beneficial mutations for strong selective pressures such as antibiotics can be identified using low-editing efficiency libraries as well. For example, rifampicin resistance mutations have been identified using high-throughput CRISPR-enabled trackable genome engineering (CREATE) technology using several gRNAs (Garst *et al*, 2017). However, in CREATE, despite using a much larger library with saturation mutagenesis of the same *rpoB* window, the study managed to identify 10 resistance-conferring single mutations at three residues, as opposed to over 41 beneficial mutations identified at 17 residues identified with CREPE. Additionally, as opposed to any previous study, using the sensitive fitness estimates from CREPE, data collected from 1 week of experiments corroborated biochemical, evolutionary, and epidemiological findings gathered over two decades of previous research. These included mapping of rifampicin interacting and clinically significant residues (Fig 6B and C), the biochemical basis of increased resistance paths at lower concentration (Figs 6B and EV4), and the effect of rifampicin dosage on favoring different spectrum of mutations (Figs 6D and EV4). Additionally, unlike previous high-throughput studies, using the double mutants obtained using the error-prone PCR template, we also managed to understand the trends in epistasis and evaluated the impact of compensatory mutations on improving fitness (Fig 7), which plays a significant epidemiological role in transmission of drug-resistant *M. tuberculosis*. To the best of our knowledge, no other technology to date has demonstrated the fitness estimation for an essential gene at such high resolution. Such expedited DMS of essential genes can be an invaluable tool for rapid understanding of their function in diverse cellular processes.

The higher efficiency, diversity, and better fitness estimates with CREPE as opposed to other high-throughput CRISPR-mediated technologies are due to the use of a single active gRNA, which ensures that all cells during the library construction undergo the same amount of DNA DSBs and helps to avoid biases in the library construction. In addition, each mutant in the library has the same synonymous PAM mutation, further controlling sources of noise in fitness estimates. Using a single SPM also assures both high-efficiency editing and minimal fitness effects associated with the background SPM. The diversity can be generated in a single transformation step. Finally, we directly sequenced the genomic loci to track the variants, which allow direct fitness estimates without the need to use barcodes as a proxy.

In its current format, there are several limitations of using CREPE. Firstly, the technology is the limited target size (300–400 bp). In order to overcome this limitation, we will have to use a tiling approach to study entire proteins using shorter windows. We will need to identify functional gRNA for each tile. Therefore, studies using CREPE were limited to only particular genomic regions. For genome-wide studies, previous high-throughput and multiplexed platforms are still the only viable approach. Secondly, using error-prone PCR to generate libraries only generates single nucleotide changes per codon. Therefore, for each amino acid a maximum of 11 other amino acid substitutions are accessible. Several previously known mutations such as the clinically relevant mutation S531L were not identified in our study because they require two consecutive nucleotide substitutions. This mutation has been identified in *M. tuberculosis*, which has a different codon usage than *E. coli*. Therefore, while this mutation is accessible by a single nucleotide change in *M. tuberculosis*, it would require two consecutive nucleotide changes in *E. coli*. Such changes are hard to achieve using error-prone PCR. Additionally, previous studies have highlighted that 2–3 consequent nucleotide changes, to enable complete saturation mutagenesis to more distant amino acids, often lead to more diverse chemical changes that have larger fitness effects (Pines *et al*, 2015; Garst *et al*, 2017). However, CREPE is amenable to any plasmid mutagenesis technique and so site saturation can be achieved using the same workflow. Therefore, in further iterations of CREPE can we aim to address current limitations to expand the scope of the CREPE technology.

# Materials and Methods

### Reagents and Tools table

| Reagent/Resource | Reference or source | Identifier or catalog number |
|---|---|---|
| **Experimental models** | | |
| *Escherichia coli* (MG1655) | | |
| **Recombinant DNA** | | |
| Cas9-lambda Red recombination dual vector | Copley Lab (preprint: Morgenthaler *et al*, 2019) | pAM053 |
| gRNA expressing plasmid | Addgene | Cat# 71656 |
| Error prone PCR library contruction bacobone (pSAH031) | Addgene | Cat#90330 |
| portMAGE plasmid for mutL-E32K insert | Addgene | Cat#72677 |

**Reagents and Tools table**  (continued)

| Reagent/Resource | Reference or source | Identifier or catalog number |
|---|---|---|
| **Oligonucleotides and sequence-based reagents** | | |
| PCR primers | This study | Appendix Table S2 |
| Homology Arm templates | This study | Appendix Table S2 |
| gRNA spacer sequences | This study | Appendix Table S2 |
| **Chemicals, enzymes and other reagents** | | |
| E. Cloni 10G Elite Electrocompetent Cells | Lucigen | Cat# 60052-4 |
| LB Agar | VWR | Cat# L9115 |
| OneTaq 2X MasterMix | NEB | Cat# M0482L |
| Q5 polymerase | NEB | Cat# M0492L |
| NEBuilder | NEB | Cat# E2621L |
| DpnI | NEB | Cat# R0176 |
| QIAquick Gel Extraction Kit (250) | Qiagen | Cat# 28706 |
| QIAprep Spin Miniprep Kit (250) | Qiagen | Cat# 27106 |
| QIAquick PCR Purification Kit (250) | Qiagen | Cat# 28104 |
| Spectinomycin | Fisher | Cat# 50653816 |
| Chloramphenicol | Fisher | Cat# AAB2084122 |
| Rifampicin | Fisher | CAT# BP2679250 |
| Random mutagensis genemorph II Kit | Agilent | Cat# 200552 |
| **Software** | | |
| Usearch | Edgar (2010) | |
| Benchling | https://benchling.com | |
| Python 2.7 | http://www.python.org | |
| Biopython | https://biopython.org/ | |
| **Other** | | |
| Illumina Miseq 2X250 | Illumina | |

## Methods and Protocols

### Strains and plasmids and cloning methods

All editing experiments were performed in *E. coli* K-12 MG1655 strain. The *E. coli* was grown in lysogeny broth (LB) media for all experiments.

For our initial experiments, we used the pAM053 plasmids previously published (preprint: Morgenthaler *et al*, 2019), encoding *cas9* expressed under the weak pro1 promoter (Davis *et al*, 2011) and lambda Red recombination proteins expressed under the lambda phage pL promoter, controlled by the temperature-sensitive cI857 repressor. The *cas9*+lambda Red recombination+*mutL-E32K plasmid* was constructed by cloning the *mutL-E32K* gene in the lambda Red recombination operon of pSIM5 using the primers described previously (Nyerges *et al*, 2016). This allowed expression of *mutL-E32K* using the temperature-inducible pL promoter as well, such that the MMR machinery was blocked only during recombineering. The mutL-E32K was amplified using the pORTMAGE-2 plasmid (https://www.addgene.org/72677/). The pORTMAGE-2 was a gift from Csaba Pál (Addgene plasmid # 72677; http://n2t.net/addgene:72677; RRID:Addgene_72677). The gRNA plasmid expressed under the J23119 promoter was purchased from Addgene (https://www.addgene.org/71656/). For

each of the genome targets, different spacers were cloned in the gRNA for the different targets (Appendix Table S3). In order to construct the error-prone PCR libraries, the target with end homologies (500 bp for *galK* and 250 bp for *crp*, *rpoB*, and *mreB*) on each side was amplified from the genome and cloned into the pSAH031 backbone (https://www.addgene.org/90330/), which was recently developed for construction of unbiased plasmid mutagenesis libraries (Higgins *et al*, 2017). The pSAH031 was a gift from David Savage (Addgene plasmid # 90330; http://n2t.net/addgene:90330; RRID:Addgene_90330). Subsequently, 250-bp-long gblocks with the synonymous PAM mutation (SPM) were used to replace the wild-type sequence in the error-prone plasmids using CPEC cloning (Quan & Tian, 2011). All amplifications were performed using Kapa Biosystems high-fidelity polymerase (catalog #07958897001) following the manufacturer-defined amplification protocol. The plasmid inserts and backbones were amplified with sufficient overlapping homology (35–40 bp long), and all cloning was performed using circular polymerase extension cloning or CPEC (Quan & Tian, 2011) using 12.5 μl (with at least 100 ng of backbone) of an equimolar insert: backbone mixture, and 12.5 μl of NEB 2× Phusion Master Mix (catalog #M0530). PCRs were performed using the following cycling conditions: initial denaturation at 98°C-30 s, 10× (98°C-10 s, 55°C-10 s, 72°C-90 s) and a

final extension at 72°C for 120 s followed by a hold at 12°C). 10 µl of the CPEC reaction was transformed into competent cells by electroporation. The transformed cells were plated on LB and appropriate antibiotics as listed in the table, and colonies were sequence-verified.

### Cas9-mediated lambda Red recombineering

Cas9-mediated lambda Red recombineering was used to introduce single and a library of edits on the genome following the heat-shock protocol (Sharan *et al*, 2009). The homology donor templates with single edits or error-prone libraries were amplified and gel-purified with 250-bp end-homology arm length unless described otherwise in the text. Starter cultures of cells with the plasmids encoding *cas9* and lambda Red recombination or plasmids encoding *cas9*, lambda Red recombination, and *mutL-E32K* were grown overnight at 30°C. In the morning, fresh cultures were started with a 100-fold dilution of the overnight cultures in fresh media and grown at 30°C up to a mid-log OD measured at 600 nm of 0.4–0.5. At this point, the cells were immediately placed in a shaking water bath set at 42°C and the lambda Red recombination operon (and MutL-E32K in some cases) was induced for 15 min. Then, the cells were immediately placed on ice and rapidly cooled by shaking and kept on ice for 15 min. The cells were centrifuged at $7,500 \times g$ at 4°C for 3 minutes. The pellet was washed with 25 ml ice-cold 10% glycerol solution by resuspending and centrifuging at $7,500 \times g$ at 4°C for 3 min thrice. Finally, the cells were resuspended at 100-fold lower volume of ice-cold 10% glycerol than the starting culture volume. Dialyzed mixtures of the gRNA-expressing plasmid and single donors with point mutations or library of mutations were electroporated at 1.8 kV into 50 µl of the washed now electrocompetent cells. The cells were recovered in LB for 3 h at 30°C and subsequently plated on LB+agar and appropriate antibiotics.

### CREPE: Screening the target gRNA and spacer

1  Use the *E. coli* reference genome for MG1655 (accession number U00096) to get the sequence for the target of interest for mutagenesis. Target regions will be 300–400 bp long. Within the target sequence, identify the most centrally located PAM (5′-NGG-3′) sequence. The 20-bp sequence upstream of the PAM sequence is the N20 spacer sequence in the gRNA (Jinek *et al*, 2012). There are usually multiple possible PAM/spacer combinations close to the center of the target. Use guide scoring algorithm tool on the Benchling website (www.benchling.com) to start with the gRNA with the highest score.

2  *Screening the gRNA for activity*: Clone the gRNA spacer into the gRNA-expressing plasmid backbone (https://www.addgene.org/71656/). Then, test the Cas9:gRNA-induced cell death associated with the gRNA as follows. In two independent transformations, transform the gRNA targeting the region of interest and a non-targeting gRNA into *E. coli* cells with the Cas9-lambda Red recombination dual plasmid. Subsequently, plate several dilutions of the transformations on LB Agar plates with ampicillin (the resistance marker for the gRNA plasmid) and chloramphenicol (the resistance marker for the Cas9-lambda Red recombination dual plasmid). Grow the cells overnight at 30°C. The non-targeting gRNA does not induce a DSB on the genome and therefore does not lead to any Cas9:gRNA-induced

cell death. A targeting gRNA that causes more than three orders of magnitude reduction in the number of CFUs per transformation compared with the non-targeting gRNA due to Cas9:gRNA-induced cell death will be sufficient for successful editing using Cas9:gRNA-mediated recombineering.

3  *Screening the gRNA for editing*: Clone the target into the pSAH031 backbone along with sufficient non-mutated homology arms on each site (~ 250–300 bp's). Use standard site-directed mutagenesis to introduce the synonymous PAM mutation (SPM) for the PAM corresponding to the most active screened gRNA. Amplify the target + homology arm + SPM template and purify the PCR product by gel extraction. Following the Cas9-mediated lambda Red recombineering protocol, make competent cells with active Cas9 and lambda Red recombination and co-transform 100 ng of the purified PCR product with 100 ng of the gRNA-expressing plasmid. Recover the cells for 3 h and then plate the cells on ampicillin (the resistance marker for the gRNA plasmid) and chloramphenicol (the resistance marker for the Cas9-lambda Red recombination dual plasmid). Use colony PCR to amplify the mutated region from the genome and sequence 10–20 edited colonies. A good gRNA + SPM template combination should enable > 98% efficiency for incorporation of the SPM on the genome.

### CREPE: Error-prone PCR library construction

1  The error-prone PCR libraries were constructed using the Agilent GeneMorph II Random Mutagenesis Kit (Part #200550). Design primers for the targeted region with overlaps to allow cloning into the pSAH031 backbone where we cloned the target with the unmutated homology arms in the previous step (Appendix Table S3).

2  The mutation rate in the PCR can be controlled by altering the initial template amounts. To obtain high and low diversity, use 10 ng and 400 ng of template plasmid for the error-prone PCR, respectively.

3  Follow the Agilent GeneMorph II™ instruction manual to set up the PCR. Use the following PCR cycling conditions: initial denaturation: 95°C-2 min, 30× (95°C-30 s, Tm-30 s, 72°C-1 min), and final extension: 72°C-1 min. Determine the PCR Tm using NEB's Tm calculator for Taq polymerase.

4  After error-prone PCR, add 1–2 µl of NEB's DpnI directly to the PCR to digest the template plasmid used for the PCR and incubate the PCR + DpnI mixture for 2 h at 37°C.

5  Purify the libraries using Qiagen's gel purification kit.

6  Clone the purified PCR product into their respective backbones with unmutated end homologies using the NEBuilder HiFi DNA assembly kit (catalog #E2621) using the manufacturer guidelines for reaction setup.

7  Transform the clones into Lucigen Elite E. cloni electrocompetent cells (catalog #60061) and plate several dilutions of the transformation reaction on LB agar + kanamycin (the antibiotic marker for pSAH03) and grow overnight at 37°C. Make sure to plate several dilutions of the cells to ensure that the library can be isolated from plates with well-resolved colonies.

8  For each error-prone PCR experiment, collect 50,000–100,000 colonies by scraping several plates in liquid LB.

9  Extract the plasmid-based error-prone library using the Qiagen Miniprep extraction columns.

### CREPE: Genomic library construction

1  Design primers with the desired homology arm length to amplify the error-prone donor library from the error-prone plasmid library.

2  Amplify the donor library. In this amplification step, one needs to ensure that the libraries are not overamplified to avoid biases. In order to avoid overamplification, use high initial template amounts (~ 10 ng per PCR) and fewer PCR cycles (13–20 amplification cycles). The number of cycles must be optimized by attempting the PCR with varying numbers of cycles. Proceed using the minimum number of cycles that is able to produce a visible band on an agarose gel. Perform the PCR with a high-fidelity polymerase such as Kapa polymerase following the manufacturer instructions for set up.

3  After amplification of the donor library with the optimized conditions, use Qiagen's PCR purification kit to purify the DNA sample. In order to achieve high concentrations, perform 4–8 PCR replicates and purify on one column.

4  After PCR purification, digest the template plasmid used for the PCR: mix 43 μl of the PCR + 2 μl of NEB DpnI + 5 μl of 10× CutSmart buffer. Incubate the PCR + dpnI mixture for 2 h at 37°C.

5  Purify the DpnI-digested PCR product using the Qiagen gel purification kit.

6  Following the Cas9-mediated lambda Red recombineering protocol, make competent cells with active Cas9 and lambda Red recombination. Co-transform 100 ng of the purified PCR product with 100 ng of the gRNA-expressing plasmid into the cells. Recover the cells for 3 h at 30°C.

7  Plate several dilutions of the recovered cells on LB agar plates with ampicillin (the resistance marker for the gRNA plasmid) and chloramphenicol (the resistance marker for the Cas9-lambda Red recombination dual plasmid). Grow the plates at 37°C.

8  Depending on the expected library diversity, scrape 20–50× more colonies than the expected diversity to obtain adequate coverage of the libraries.

### Rifampicin selection

As described above, we constructed two independent libraries of genomic error-prone PCR-mediated *rpoB* libraries using the CREPE protocol. For each library, we scraped ~ 10,000–15,000 colonies after resuspension in LB and stored multiple glycerol stocks of the library. We also scraped ~ 10,000–15,000 colonies obtained from cells transformed with the gRNA + dsDNA homology donor with only the SPM. Prior to the selection, we thawed one glycerol stock of library 1, two glycerol stocks of library 2, and three glycerol stocks of the gRNA + SPM only and grew them independently in 100 ml LB at 37°C for 4 h to cure the plasmids encoding *cas9*, lambda Red recombination proteins, and *mutL-E32K*. Simultaneously, we diluted overnight cultures of single isogenic cultures of wild-type *E. coli* MG1655 parent and *E. coli* MG1655 + SPM only in LB and grew them at 37°C for 4 h. After growth, we normalized the OD measured at 600 nm for each of our samples and plated 100 μl of several dilutions of the cultures on three different concentrations of rifampicin. We used the plates with the most well-resolved colonies to determine the resistant CFUs/ml of rifampicin. We saved the glycerol stocks used for each time point for the zero time point ($t_0$) of the selection experiment, and we scraped ~ 10,000–15,000 at each concentration as the final time point ($t_t$) of the selection experiment for next-generation sequencing.

### CREPE: Genomic deep sequencing

1  Extract DNA using the boiling protocol. Wash 50 μl of cell sample for each experiment twice with PBS. In order to wash, centrifuge the cells at 7,500 *g* and then resuspend the cells in 1 ml PBS. Finally, resuspend the pellet in 50 μl of TE buffer (pH 8.0). Boil the resuspended cells at 100°C for 10 min.

2  Amplify each library using primers with Illumina Nextera adapters (Appendix Table S3) using Kapa polymerase following manufacturer guidelines using 1 μl of the cell extract. It is important to avoid overamplification for next-generation sequencing. Refer to the CREPE: genome library construction step for the protocol for optimizing cycles to prevent overamplification.

3  Amplify the PCR product for each experiment with a unique experimental Nextera barcode. Paired-end 2*300-bp read sequencing was performed using the Miseq platform following manufacturer guidelines for sequencing.

### CREPE: Data analysis

1  *Next-generation sequencing data analysis*: A custom analysis pipeline was built to quantify recombination trends and fitness calculation for our experiments (https://github.com/Alaksh/CREPE-Analysis-Code). Paired Illumina reads were first assembled using the Usearch Mergepairs algorithm (Edgar, 2010). Then, the assembled reads were aligned to the wild-type sequence of the appropriate target gene using the usearch global alignment algorithm (Edgar, 2010). The alignment generates a text file which summarizes details (id, number of mutations, *E*-value, mismatches, indels, and mismatch/indel positions as a qrowdot alignment output). Variant counts were estimated as value counts for the qrowdot alignment output. Finally, we wrote a custom code to extract information about the nucleotide and amino acid changes corresponding to each qrowdot output.

2  *Fitness calculations for resistance to rifampicin*: Sample code for calculating the resistance to rifampicin has been provided (https://github.com/Alaksh/CREPE-Analysis-Code). Fitness calculations for resistance to rifampicin with each replicate were estimated using the two time point enrichment score calculation algorithm described previously (Rubin *et al*, 2017). The fitness for each variant was estimated as follows:

$$\text{fitness, } f = \log\left(\frac{C_{i,sel} + 0.5}{C_{wt,sel} + 0.5}\right) - \log\left(\frac{C_{i,input} + 0.5}{C_{wt,input} + 0.5}\right)$$

where $C_i$ is the total count for a variant "i" in the library and $C_{wt}$ is the total count for the wild-type reference in the library. "Sel" signifies the counts obtained after selection on rifampicin, and "input" signifies counts before selection. For each score, we estimated an error using a Poisson approximation.

$$\text{Standard Error} = \text{sqrt}\left(\frac{1}{C_{i,sel}} + \frac{1}{C_{wt,sel}} + \frac{1}{C_{i,input}} + \frac{1}{C_{i,input}}\right)$$

After fitness estimates, we filtered the data to eliminate reads that may be erroneous. Since we targeted an essential gene, the

occurrence of stop codons would be impossible. Therefore, we eliminated reads with the following filter:

$$C_i \geq C_{max-stopcodon} + 2.56 * \text{sqrt}(C_{max-stopcodon})$$

After read filtering, we combined the fitness estimates from each replicate using maximum-likelihood estimates for variant score and standard error using Fisher scoring iterations (Rubin *et al*, 2017). The same protocol was also repeated with only synonymous mutations in our dataset. We classified a mutant as resistant to rifampicin if the fitness was 2.96 standard deviations greater than the mean fitness of synonymous mutations.

### Epistasis measurement

We first isolated the double mutants for which we had fitness scores for the single mutants. We then calculated the sum of fitness scores and the error associated with the sum using standard propagation of error. In order to assign magnitude epistasis, we used the following definition:

$f_{AB} > f_A + f_B$: positive
$f_{AB} = f_A + f_B$: neutral
$f_{AB} < f_A + f_B$: negative

where $f_A$, $f_B$, and $f_{AB}$ are fitness values for mutant A, mutant B, and combine mutant AB. In order to statistically determine the magnitude of epistasis, we performed a one-tailed Student's *t*-test using the fitness and standard error in estimation of the fitness, where the null hypothesis that the sum of fitness is the same as the fitness of double mutant was rejected only when the *P*-value was $\leq 0.01$.

## Data availability

The data and code produced in this study are available in the following databases:

- Deep sequencing data: NCBI Gene Expression Omnibus GSE143629 (http://www.ncbi.nlm.nih.gov/geo/query/acc.cgi?acc =GSE143629).
- Code: GitHub (https://github.com/Alaksh/CREPE-Analysis-Code).

**Expanded View** for this article is available online.

### Acknowledgements

We would like to acknowledge the University of Colorado Boulder's next-generation sequencing core headed by Dr. Amber S. Scott for support with deep sequencing for our study. We would also like to thank Dr. Carrie Eckert, Andrew Mogenthaler, and Dr. Shelley Copley for sharing plasmids. This work was supported by the US Department of Energy DE-SC0018368. AC was also funded by the 2017–2018 STEM Chateaubriand Fellowship sponsored by the French Institut National de la Santé et la Recherche Médicale (INSERM) European Research Council, FP7 grant 310944 to O.T.

### Author contributions

Conceptualization: AC, OT, and RTG. Investigation: AC, JAF, and RGF. Formal analysis: AC and OT. Visualization: AC and JLK. Writing—original draft: AC, OT, and RTG. Writing—review and editing: AC, JAF, RGF, JLK, OT, and RTG. Supervision: JLK, OT, and RTG. Funding acquisition: RTG and OT.

### Conflict of interest

The authors declare that they have no conflict of interest.

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
