## [Review Process File · Molecular Systems Biology]

CRISPR/Cas9 recombineering-mediated deep mutational scanning of essential genes in *Escherichia coli*

Alaksh Choudhury, Jacob Fenster, Reilly Fankhauser, Joel Kaar, Olivier Tenaillon, and Ryan Gill

Review timeline:	Submission date:	26th September 2019
	Editorial Decision:	6th November 2019
	Revision received:	16th January 2020
	Editorial Decision:	10th February 2020
	Revision received:	12th February 2020
	Accepted:	13th February 2020

Editor: Maria Polychronidou

Transaction Report:

1st Editorial Decision

6th November 2019

Thank you again for submitting your work to Molecular Systems Biology. We have now heard back from the three referees who agreed to evaluate your study. As you will see below, the reviewers acknowledge that the presented method seems relevant for the field. They raise however a series of concerns, which we would ask you to address in a major revision.

The reviewers' recommendations are rather clear and therefore I see no need to repeat the points listed below. Please feel free to contact me in case you would like to discuss in further detail any of the issues brought up by the reviewers.

REFeree REPORTS

Reviewer #1:

In this study, Choudhury et al developed a high-throughput method for mutating genes, including essential genes, in their native genetic context in *E. coli*, named CRISPR/Cas9-mediated genomic Error-Prone Editing (CREPE). By using longer homology arms and blocking mismatch repair, the efficiency of CREPE was improved to 85% for non-essential genes and 55% for essential genes. Application of CREPE on *rpoB* combined with next generation sequencing for deep mutational scanning led to better understanding of resistance to rifampicin. This serves as a nice example to show deep mutational scanning of antibiotic targets is a powerful tool for exploring mechanism of antibiotic resistance and further understanding the resistance evolution process. In summary, CREPE is a nice tool with some limitations for high resolution study of protein function, which can provide detailed landscape of protein function, and mechanistic insights into protein structure and activity. We believe this is a nice tool for the bacterial field and will promote research on function of

essential genes, antibiotic resistance related genes, also other genes by incorporating with appropriate report systems.

We have several minor and major comments on the manuscript.

Major comments

Figure 2D. Missing the mutation frequency per base in the donor DNA library. Without this data, you cannot conclude that bias is caused by transformation. You have this data as you used it in Figure 2B. Just do further analysis and include it in Figure 2D.

Figure 2E. The mutation efficiency of donor DNAs with 50, 150 and 250 bp should be shown here, to exclude the difference of mutation efficiency caused by difference on mutation efficiency of donor DNAs. Same for Figure EV1D.

L165-169, conclusion is not solid here. 1) There is no comparison of mutation distribution of the low diversity and high diversity donor DNAs. Although it is mentioned in the figure legend that high diversity has 3-4 mutations and low diversity has 1-2 mutations on average, it would still be nice to show this with a format like Figure 1C. 2) A good confirmation study to conclude on this can be done like this: use two classes of templates, one with 3-4 mutations ; the other with 1-2 mutations to measure recombination efficiency with different length of HAs, like 50 bp and 250 bp.

In addition, as for a technical tool, it is always better to have higher efficiency, here you showed that with longer HA of 250 bp, you can improve efficiency. How about even longer? Maybe not necessary to repeat the whole process but include longer HAs in the above suggested 2) confirmation study to let reader know whether there is possibility to improve the efficiency further.

Line 239-241, Figure 4C. "we observed a significant increase in mutation frequency per base across the target region", significant increase compared to what? The Figure 4C shows bias to variants with mutations close to SPM for *crp* and *mreB*. Figure 4D, the meaning of x-axis is not well explained, which makes it difficult to understand the conclusion of unbiased distribution.

Minor comments

L57 and L182: Red to red

L134: the genome the genome -> the genome

L155: For Figure 2E. 11.4 +/- 0.2%, how did you calculate? Is it not 8.3 +/-1.4?

L200: MutL-E32K -> MutL-E32K

Fig. 3a: Promoter (not Prmoter)

Figure 3C: Add the donor DNA information of the libraries with 50 and 250 bp HA

Reviewer #2:

The authors present an improved genome editing technology to generate targeted genomic mutation libraries in *E. coli*, combining CRISPR-Cas9 and recombineering systems. Repair template variant libraries, containing a mutated synonymous PAM sequence, are generated using error-prone PCR and are flanked by non-mutated homology arms. These are introduced into cells along with a plasmid expressing a single guide RNA. The strain also contained a plasmid expressing Cas9 and the lambda red recombineering machinery. The authors perform proof-of-principle experiments on a portion of the *galK* gene, demonstrating that they can mutate this region using this system, albeit with biases in mutation number per individual sequence and mutation position relative to PAM. The biases are partially alleviated by increasing the homology length of the repair template and by expressing a dominant negative allele of the mismatch repair protein MutL. They then use the developed system to generate positional mutations in the essential gene *rpoB* and identify both novel and already-identified mutations that confer resistance to rifampicin. The data both corroborated with previously established models of resistance and improved upon them considerably.

The presented work is novel and a considerable improvement on current methods of bacterial deep mutational scanning. Variant effect mapping is becoming a field with ever increasing significance, and this work would be of interest to this community. Some elements of the work would benefit from additional analyses to strengthen claims made by the authors. Specific comments follow.

Comments:

Lines 107-111: the lambda red machinery is expressed using a heat inducible promoter (induced at 42 degree C) and contains a curable origin of replication that is cured from the cell at 37 degree C. Please briefly elaborate on how expression of the lambda red proteins is achieved while not losing the plasmid.

Lines 164-169: Section is unclear, especially by what is meant by "improving recombination with more divergent sequences in the genome". Is the definition of divergent herein those sequences with many mutations? Please reword to more clearly define.

Figure 2G: The shown graphic doesn't clearly demonstrate that an increase in homology arm length results in a higher frequency of high diversity sequences in the genome. Can a statistical measure be employed to validate the statement?

Lines 186-187: Graphical demonstration that expression of MutL-E32K significantly decreases PAM-proximal positional bias is needed.

Line 197: Brief elaboration on the nature of the typical mechanism of action of MutL would be helpful here.

Lines 204-227: The authors may want to consider removing this section. The authors describe a compelling model, where intermediates from template switching are repaired by RecA-mediated strand invasion (which is de-repressed by expression of MutL-E32K), but its description may be better suited in a separate publication, as it does not add significant value to this work.

Lines 234-239 and Figure 2B: A comparison of the diversity and composition of the pre-transformation donor libraries for each gene (in terms of number of mutations per sequence) is needed to compare the resulting mutation efficiencies of each. Moreover, some additional analyses can be performed to strengthen the suggestion that the mutation frequency is lower in essential genes due to the creation of loss of function alleles that don't survive to sequencing. A suggestion is to compare the diversity of codons (synonymous vs stop codons) between libraries and resulting bacterial sequences.

Lines 240-241 and Figures 4D, 4C: The claim that there is an unbiased distribution of variants is not supported with Figure 4C and is unclear in Figure 4D. I'm afraid that this claim may be overreaching, especially if compared to other methods of variant generation.

Lines 259-260: why were these concentrations of rifampicin chosen?

Lines 265-266 and Figure 5D: To further support the claim that the resulting sequenced mutations after selection reflected on the diversity of the library, it is recommended that a no-drug control (also grown for the same number of doublings as the treatment samples) be included to account for non-library mutation selection.

Line 344 (paragraph): the findings are interesting and would benefit from restructuring of the paragraph. An initial comment on a "trade-off" between the potential detrimental effects of mutating of an essential gene vs. the fitness advantage of mutation for drug resistance would be helpful here. Line 355: how were the 8 mutants chosen for the analysis shown in EV-3D? Can the fitness scores of additional resistant mutants be quantified from no drug pools? Please consider moving this to a main figure.

Epistasis analysis and Figure 7: How were the shown mutants selected? In Figure 7A, are the epistatic relationships statistically significant? The fraction of mutations with epistatic relationships is exceptionally high and would benefit from a thorough statistical analysis. Figure 7C and 7D do not add significant information - please consider removing.

Minor comments:

Lines 185-187: Refer to figure 2E to illustrate that expression of MutL-E32K increased the

maximum mutation frequency.

Lines 337-338: consider rewording

Line 702: remove first "to"

Line 743-744: Repeated from previous figure description.

Line 560: Kapa (not Kappa)

Lines 562-563: consider rewording sentence (Paired end 2*300bp read sequencing was performed using the Miseq platform).

Figure 5E: description of black line missing

Reviewer #3:

The authors present somewhat incremental advances in recombineering-based library construction at endogenous loci of *E. coli*, with a particular focus on conducting "deep mutational scans" of essential genes. While the tweaks that improved the experiment technically were previously published (e.g. increasing homology arm length, dominant negative inhibition of *mutL*), the authors offer, in addition to their method, an informative, high-quality dataset examining the effects of many mutations in two regions of *rpoB*. Thus, the method seems like a valuable tool for the mutation of essential genes in *E. coli*. The target audience of the advances presented in the manuscript may be limited to those focused on genome engineering in *E. coli*; while this fact dampens my own enthusiasm, the work's overall focus on a method to improve genome engineering seems appropriate for MSB.

Major concerns

My understanding of a deep mutational scan is an experiment that characterizes the majority of possible beneficial, neutral, and deleterious mutants with equal depth. However, in the cases presented here, many mutations integrated at the essential gene are immediately lost in the population due to their deleterious effects on gene function. This loss accounts for the reduced "editing efficiency" at essential vs. non-essential genes (~50% vs 80%). It would be useful for the authors to characterize those variants that are integrated and subsequently lost as lethal based on their frequency in the donor library. The input frequency of each variant should allow the experimenters to define an expectation that conservatively separates chance drop-outs from genuinely deleterious variants.

Related to the above comment, stop codons are used to filter the data, but the reader is not provided with a representation of how deleterious the stop codons are in these data and these data ought to be shown. I generally agree with the statement in the methods that within "an essential gene, the occurrence of stop codons would be impossible," but these stop-containing variants are a useful reference point in deep mutational scans, even in an essential gene.

Is there a reason that mutagenesis data from the other scanned genes are not presented? *galK*, *crp*, and *mreB* are all presented in Figure 4A, but are not revisited. A clearer explanation as to why *rpoB* is the focus would be helpful, along with at least a minimal characterization of the data of the other three genes.

The section "Alternate RecA-mediated mechanism may explain recombination using templates with single nucleotide changes" feels out of place in the Results section. The experiments in the paper may offer insight into the mechanism of action, but this mechanism is not presented clearly or convincingly enough to add weight to the rest of the paper's results. I suggest removing this section.

Minor concerns

The size of the window for efficient mutagenesis is presented variably throughout the paper. The

effective size in Figure 4 of ~200 bp seems significantly lower than the suggestion in the discussion that 300-400 bp regions are easily targeted by the method.

A window of ~250 bp was targeted for mutagenesis, but no explanation is offered for the positional variability in mutagenesis rate outside of the PAM-adjacent bias, in particular the steep drop-off near the borders.

A more useful axis would be valuable in Figures 4C. It would be more helpful to show the number of mutations per codon rather than the more abstract frequency.

Also with the target size, the paper's introduction and discussion extol the benefits of using a single, efficient gRNA, but undercut that benefit by suggesting that guides tiling across a gene will be necessary for doing more comprehensive deep mutational scans. Because this is mostly a methods paper, the authors should make a clearer case for the utility of each approach, rather than casting the CREATE and CREPE methods as hero or villain on demand.

In the discussion of the rifampicin resistant mutants, the text does not comment on an important class of mutations: those resistance mutations known to the literature and that were tested in this assay but showed no resistance. If there are none in this class, it would be worth mentioning. Information appears to be in Appendix Table 1.

In the discussion of epistasis, Figures 7A and 7C appear use different thresholds for identifying instances of negative or positive epistasis. Please clarify in the text.

From the text, it appears that negative epistasis occurs when "actual fitness was lower than the fitness-sum" (lines 393-394). Does this mean that the threshold was simply any value in which the deviation from the fitness-sum was < 0 ? The methods section indicates that some statistical cutoff was used, but this isn't apparent in the results text or figure legend.

I was confused by the statement on lines 453-454 that "using error prone PCR to generate libraries only generates single nucleotide changes." This study contains plenty of templates that had more than a single nucleotide change, but I believe the authors refer to consecutive mutation events on a single template. This limitation would prevent access to certain codons that require more than a single nucleotide change.

1st Revision - authors' response

16th January 2020

Response to Reviewers:

#Reviewer 1:

Major comments

Figure 2D. Missing the mutation frequency per base in the donor DNA library. Without this data, you cannot conclude that bias is caused by transformation. You have this data as you used it in Figure 2B. Just do further analysis and include it in Figure 2D.

The reviewer raises a valid point on the mutation distribution. As a response, we have added the data in Figure 2D and 2F to present the distribution of mutations in the donor. We also added a sentence in the text to highlight unbiased distribution in the donor as:

Line 148-149: "While the mutation frequency in the donor was consistently high, we observed a decrease in the mutation frequency per residue with increasing distance from PAM on the genome (**Figure 2D** and **Figure EV 1C**)."

Line 173-174: "The λ after genome integration with 50 bp long end homology with the high diversity donor was significantly higher than that of the donor library itself."

Figure 2E. The mutation efficiency of donor DNAs with 50, 150 and 250 bp should be shown here, to exclude the difference of mutation efficiency caused by difference on mutation efficiency of donor DNAs. Same for Figure EV1D.

This is an excellent point made by the reviewer that differences in the donor may lead to differences in editing efficiency. In our experiments, we amplified the donor libraries for the 50 bp, 150 bp and 250 bp long end-homology using the same error-prone PCR plasmid template, which was built using ~300 bp of end-homology in the plasmid. We made sure to amplify the DNA from the plasmid by taking care that overamplification and bias can be avoided. So, we are confident that the diversity for each template would be the same as the diversity of the plasmid donor. We did not sequence each library to save on sequencing costs. In order to explain this, we added the following in the text:

Line 162-168: “For each HA length, the donor libraries were prepared by PCR amplification using the same error prone PCR plasmid library, but with different primers to obtain different homology arm lengths. We used very high template concentrations, high-fidelity polymerase, low and same number of amplification cycles for each PCR. Therefore, the mutation efficiency of each donor library was expected to be consistent and similar to that of the plasmid error prone PCR library. Therefore, we assumed that the variation in mutation efficiency on the genome due to variation in mutation efficiency of the donor was unlikely.”

L165-169, conclusion is not solid here. 1) There is no comparison of mutation distribution of the low diversity and high diversity donor DNAs. Although it is mentioned in the figure legend that high diversity has 3-4 mutations and low diversity has 1-2 mutations on average, it would still be nice to show this with a format like Figure 1C. 2) A good confirmation study to conclude on this can be done like this: use two classes of templates, one with 3-4 mutations ; the other with 1-2 mutations to measure recombination efficiency with different length of HAs, like 50 bp and 250 bp.

- 1) We added the Figure in the Appendix to compare the distribution of mutations between the high diversity and low diversity donor. We mention it in the text as follows:

Lines 136-137: “The high diversity donor contained a mean of 3-4 mutations per donor sequence and the low diversity donor contained a mean of 1-2 mutations per donor sequence (**Appendix Figure 1**).”

In addition, we also added statistical tests to determine the significance of differences between the distributions. We also rephrased the paragraph for improved clarity, which was also mentioned by reviewer 2:

Line 178-182: “We observed a significant increase in the percentage of sequences with higher diversity (number of mutations in addition to the SPM > 2) on the genome (**Figure 2G**) (p-value for chi-squared test < 10⁻¹⁶). Increasing the HA length did not substantially reduce the PAM-proximal mutation bias but improved mutation efficiency by improving recombination of donor sequences with a higher number of mutations per sequences on the genome.”

- 2) A good confirmation study to conclude on this can be done like this: use two classes of templates, one with 3-4 mutations ; the other with 1-2 mutations to

measure recombination efficiency with different length of HAs, like 50 bp and 250 bp.

The reviewer raises a very interesting point here. As a matter of fact, the actual mechanism for improvement of recombination is now a subject for another study in our lab. In this study, we are attempting to understand recombination and limitations to recombination exactly as the reviewer mentioned. We are using single donor templates to understand how changing the length of the homology arm and blocking MutL impact the efficiency of recombination. Some preliminary data to represent the impact of each for one such template is shown below.

We used a single template with 5 mutations and performed Cas9 mediated recombineering using 50 bp HA, 250 bp HA and 250 bp HA + MutL-E32K (**Figure A, below**). In each case ~99% of the sequences have the PAM inactivating mutations. With 50 bp HA, we observe that while 99% of the sequences have the PAM inactivating mutations, the frequency of the mutations decreases as we move away from the PAM on either side. When we increase the HA length, we observe a significant increase in the mutation frequency of the mutations further away from the PAM. Similarly, when we express MutL-E32K, we again observe a significant increase in mutations further away from the PAM. Therefore, increasing HA length and expressing mutL-E32K improve incorporation of mutations further from the PAM.

In our deep-sequencing data, we observed that while the purified donor consisted of 5 mutations, there were different combinations of mutations on the genome (**Figure B, below**). With 50 bp HA, only 11.14% of the sequences contained all mutations. When we increase the HA length, we observe that ~20% of the sequences now incorporate all mutations on the genome. Finally, by expressing MutL-E32K, the number increased to 32.1%.

With this data we highlight two things:

- 1) Increase in HA length and expressing MutL-E32K indeed affect recombination.
- 2) With complex constructs, the patterns of recombination are also complex.

The 1st point addressed some of the previous comments made by the reviewer, where the role of the increased HA length was not clear. With respect to the 2nd point above, since the nature of recombination is complicated, we would need to perform additional characterization in order to explain the data with single constructs. Therefore, we believe that including this data is beyond the scope of this paper. However, we added the necessary information in this response to address the reviewer's concerns.

In addition, as for a technical tool, it is always better to have higher efficiency, here you showed that with longer HA of 250 bp, you can improve efficiency. How about even longer? Maybe not necessary to repeat the whole process but include longer HAs in the above suggested 2) confirmation study to let reader know whether there is possibility to improve the efficiency further.

This is another excellent point raised by the reviewer. As a part of the study mentioned earlier, we also wanted to understand if the effects of increasing the homology arm length and MutL-E32K expression were mutually exclusive to understand the mechanism. We added the following data in the paper as a part of our evaluation:

Line 205-213: “Next, we evaluated if the effects of increase in homology arm length and expression of MutL-E32K were mutually exclusive. If the improvement in editing through each occurred via independent mechanisms, a decrease in the HA length while conditionally expressing MutL-E32K would decrease editing efficiency. Therefore, we repeated editing with the Cas9+lambda Red recombination+MutL-E32K system using the high diversity donor with 50 bp long

HA. Interestingly, we observed no significant difference between editing using 250 bp long HA ($81.2 \pm 0.82\%$) and 50 bp long HA ($79.9 \pm 0.82\%$) (p-value = 0.1, **Appendix Figure 2**). This suggests that increase in HA length and expressing MutL-E32K are not mutually exclusive. Additionally, any limitations in recombination with 50 bp long end Homology length can be overcome by just inhibiting MutL.”

Therefore, based on the reviewer’s comments, as far as the maximum limit for increase in editing efficiency is concerned, we think that this limit is achieved by the temporary inhibition of mutL.

Line 239-241, Figure 4C. "we observed a significant increase in mutation frequency per base across the target region", significant increase compared to what?

We changed the text for this line as follows:

Line 239-240: “For each of the genes, we observed a high mutation frequency per base across the targeted region (**Figure 4A**).”

The Figure 4C shows bias to variants with mutations close to SPM for *crp* and *mreB*. Figure 4D, the meaning of x-axis is not well explained, which makes it difficult to understand the conclusion of unbiased distribution.

We agree that the data provided was not adequately explained. Based on the above recommendation made by the reviewer, we reanalyzed the data and rewrote the section to make following changes

Lines 263-274: “For successful deep mutational scanning, multiple substitution should be possible at each residue within the target for adequate sampling of the sequence space. In addition to adequate number of substitutions, there should be an adequate number of counts associated with each substitution for efficient accurate fitness estimates. Therefore, we next analyzed the average codon substitutions and position wise variant frequencies for each of the targets. For each gene library, we scraped ~20,000 colonies. We observed several substitutions at each targeted residue within each target gene (**Figure 4D**), we observed 7.75, 7.5, and 7.55 mean substitutions per codon for *crp*, *mreB* and *rpoB* respectively (**Figure 4D**). The variant counts for substitutions at each position primarily varied between 10^2 – 10^4 counts per variant (from $\sim 2 \times 10^6$ total counts) with 439, 613 and 692 median reads per variant for *crp*, *mreB* and *rpoB* respectively (**Figure 4D**). Therefore, using CREPE we were able to successfully develop mutation libraries over the entire targeted sequence space with significant counts associated with variants for fitness mapping, of several genes in *E. coli*.”

We also updated **Figure 4D**.

Minor comments

L57 and L182: Red to red

We corrected the Red to red.

L134: the genome the genome -> the genome

We changed it to “the genome”

L155: For Figure 2E. $11.4 \pm 0.2\%$, how did you calculate? Is it not $8.3 \pm 1.4\%$?

We calculated the percent change in percentage editing as $(\text{final-initial})/\text{initial} * 100 = (67.2-60.3)/60.3 * 100 = 11.44\%$. Then used standard error propagation for the error. We found that the error we reported has an error. It is 2.14%. We updated the same in the paper in Line 161.

L200: Mutl-E32K ->MutL-E32K

We changed the text to MutL-E32K

Fig. 3a: Promoter (not Prmoter)

We changed to Promoter

Figure 3C: Add the donor DNA information of the libraries with 50 and 250 bp HA

We added the donor information for all figures.

Reviewer #2:

The authors present an improved genome editing technology to generate targeted genomic mutation libraries in *E. coli*, combining CRISPR-Cas9 and recombineering systems. Repair template variant libraries, containing a mutated synonymous PAM sequence, are generated using error-prone PCR and are flanked by non-mutated homology arms. These are introduced into cells along with a plasmid expressing a single guide RNA. The strain also contained a plasmid expressing Cas9 and the lambda red recombineering machinery. The authors perform proof-of-principle experiments on a portion of the *galK* gene, demonstrating that they can mutate this region using this system, albeit with biases in mutation number per individual sequence and mutation position relative to PAM. The biases are partially alleviated by increasing the homology length of the repair template and by expressing a dominant negative allele of the mismatch repair protein MutL. They then use the developed system to generate positional mutations in the essential gene *rpoB* and identify both novel and already-identified mutations that confer resistance to rifampicin. The data both corroborated with previously established models of resistance and improved upon them considerably.

The presented work is novel and a considerable improvement on current methods of bacterial deep mutational scanning. Variant effect mapping is becoming a field with ever increasing significance, and this work would be of interest to this community. Some elements of the work would benefit from additional analyses to strengthen claims made by the authors. Specific comments follow.

Comments:

Lines 107-111: the lambda red machinery is expressed using a heat inducible promoter (induced at 42 degree C) and contains a curable origin of replication that is cured from the cell at 37 degree C. Please briefly elaborate on how expression of the lambda red proteins is achieved while not losing the plasmid.

This is an important point made by the reviewer. We have included an explanation to justify how the expression might work in the paper as follows:

Line 109-114: “Similar to several lambda Red recombination systems (Sharan et al. 2009), the plasmid also has the temperature curable pSC101 origin of replication, which replicates at 30°C and is cured from the cells at 37°C (Phillips 1999). In standard lambda red recombineering protocols, the lambda Red recombination system is induced for 15 minutes prior to recombination (Sharan et al. 2009). Since the induction time is shorter than the replication time, the plasmid should be retained in most cells.”

Lines 164-169: Section is unclear, especially by what is meant by "improving recombination with more divergent sequences in the genome". Is the definition of divergent herein those sequences with many mutations? Please reword to more clearly define.

We modified this section to include better statistical estimates and improve the arguments. We also rephrased the sentence to make it clearer:

Line 178-182: “We observed a significant increase in the percentage of sequences with higher diversity (number of mutations in addition to the SPM > 2) on the genome (**Figure 2G**) (p-value for chi-squared test < 10^{-16}). Increasing the HA length did not substantially reduce the PAM-proximal mutation bias but improved mutation efficiency by improving recombination of donor sequences with a higher number of mutations per sequence on the genome.”

Figure 2G: The shown graphic doesn't clearly demonstrate that an increase in homology arm length results in a higher frequency of high diversity sequences in the genome. Can a statistical measure be employed to validate the statement?

This is a valid point raised by the reviewer. We repeated our analyses and updated Figure 2G. In addition, we performed statistical tests to compare the two distributions. We have added the following statistical analysis in the paper:

Line 178-179: “We observed a significant increase in the percentage of sequences with higher diversity (number of mutations in addition to the SPM > 2) on the genome (**Figure 2G**) (p-value for chi-squared test < 10^{-16}).”

Lines 186-187: Graphical demonstration that expression of MutL-E32K significantly decreases PAM-proximal positional bias is needed.

We apologize that we forgot to reference the figure in this statement. We also added new statistical tests to validate the change. We added the following statement with proper references to the figures:

Line 201-204: “Expression of MutL-E32K significantly increased the maximum mutation frequency (μ , $p < 0.01$, t-test) (**Figure 3C**) and decreased the PAM-proximal positional bias of mutations (reduction in λ , $p < 0.01$, t-test) (**Figure 3C**).”

Line 197: Brief elaboration on the nature of the typical mechanism of action of MutL would be helpful here.

Lines 204-227: The authors may want to consider removing this section. The

authors describe a compelling model, where intermediates from template switching are repaired by RecA-mediated strand invasion (which is de-repressed by expression of MutL-E32K), but its description may be better suited in a separate publication, as it does not add significant value to this work.

We removed the above section and included a much-abridged version of the possible mechanism in the discussion. This version also explains the role of MutL in improving recombination:

Line 478-508: “We initially assumed that lambda Red mediated recombination using a template with limited mutations may follow a single strand intermediate dependent model proposed for dsDNA-mediated gene replacement (**Figure 2A**). Using the model, we hypothesized using a 50 bp of end homology the recombination efficiency should be similar between templates irrespective of number and position of mutations within the sequence (**Figure 1**). However, contrary to our hypotheses based on the previous model, we observed that the recombination depended on PAM-proximity and the number of mutations in the template (**Figure 2**). According to the previous model DNA polymerases is used to replicate using the mutated recombination donor as a template. Consequently mutation-specific biases (G-T, A-C, A-A, G-G > T-T, T-C, A-G >> C-C) occur due to repair by the methyl-directed mismatch repair machinery after recombination (Modrich 1991; Lahue and Modrich 1988). However, using a dsDNA with point mutations as recombination template we did not observe such mutation-biases on the genome (**Figure 3**). We think that an alternate template-switching model proposed in a recent review by Murphy (Kenan C. Murphy 2016) may explain these observations for recombineering using a template with limited mutations. According to this model, as a final transfer step the native RecA recombination machinery of *E. coli* recombines a broken chromosome with the mutations of the donor template to an intact chromosome (Kenan C. Murphy 2016). RecA-mediated recombination can form crossover products between homologous sequences within as few as 8 bp long sequences (Hsieh, Camerini-Otero, and Camerini-Otero 1992). The cells only need synonymous PAM mutations to prevent Cas9:gRNA induced cell death. So, as the distance between the SPM and targeted mutation increases, the chances of decoupling of the target mutations and the SPM increases to introduce a PAM-proximal bias (**Figure 2**). According to this mechanism, the replication machinery uses the recombination donor as a template for replication due to which the mutations in the donor are not identified as mismatches (Kenan C. Murphy 2016). This could explain why the mutation bias due to MMR are not observed. MutS and MutL block branch migration during RecA recombination (Worth et al. 1994; Tham et al. 2013). The inhibition of RecA recombination by MutL is stronger for recombination templates with a higher number of mismatches (Tham et al. 2013). This could explain the drop in recombination efficiency with an increase in the number of mutations per sequences (**Figure 2**). Additionally, this may also explain the increase in recombination efficiency with more diverse sequences when we express MutL-E32K to inhibit MutL (**Figure 3**). Lastly, RecA mediated recombination is more sensitive to HA length over 50 bp than lambda Red mediated recombination, which explains the improvement in recombination with increase in HA length over 50 bp (**Figure 2**). Further investigation of this model is needed to identify additional targets and strategies to improve CREPE.”

Lines 234-239 and Figure 4B: A comparison of the diversity and composition of the pre-transformation donor libraries for each gene (in terms of number of mutations per sequence) is needed to compare the resulting mutation efficiencies of each. Moreover, some additional analyses can be performed to strengthen the suggestion that the mutation frequency is lower in essential genes due to the creation of loss of function alleles that don't survive to sequencing. A suggestion is to compare the diversity of codons (synonymous vs stop codons) between libraries and resulting bacterial sequences.

This is again a great recommendation by the reviewer. We did observe a difference between the *galK* and *crp*, *mreB* and *rpoB* donor libraries. We reanalyzed the data and updated figure 4 to reflect this data. Please see the updated text below:

Line 239-262: “For each of the target genes, we observed a high mutation frequency per base across the targeted region (**Figure 4A**). However, the mutation efficiency for these targets was significantly lower than *galK* (**Figure 4A**). We observed a higher percentage of variants with only the synonymous PAM mutation in the donor libraries for these targets, which could explain the lower mutation efficiency in these targets compared to *galK* (**Figure 4A**). We used the same error-prone PCR protocol for each of these targets as *galK* (**Figure 4B**). Since the mutation frequency using error-prone PCR is directly proportional to amplicon length, the mutation efficiency was lower in the donors for the new targets compared to *galK*.

However, between these targets, the mutation efficiency for *rpoB* and *mreB* was significantly lower than that of *crp*, even though they had the same target size. The lower editing could be due to the essentiality of *rpoB* and *mreB*. Therefore, deleterious mutations would not be tolerated on the genome. We expected that sequences with higher number of mutations would deplete for essential genes because a higher number of mutations are more likely to be deleterious. Therefore, we compared the distribution of variants based on number of mutations per sequence (**Figure 4B**). While the distribution was comparable in the donor, we observed fewer sequences with greater than two mutations in addition to the PAM in *rpoB* and *mreB* compared to *crp* (**Figure 4B**). To further validate our hypothesis, we also compared the percentage type of mutation (synonymous, non-synonymous and stop codons) between the donor and the genome. For *rpoB* and *mreB* the percentage of synonymous substitutions increased, the percentage of non-synonymous substitutions decreased, and substitutions leading to stop codons diminished (**Figure 4C**). This observation is expected as non-synonymous mutations are more likely to have a functional impact than synonymous mutations and stop codons in essential genes would be deleterious. Few sequences with stop codons occurred in *rpoB* and *mreB* likely due to sequencing and PCR errors (**Figure 4C**). Therefore, although we obtained high mutation efficiency for essential genes, the essentiality of these genes impacted the spectrum and distribution of mutations.”

Lines 240-241 and Figures 4D, 4C: The claim that there is an unbiased distribution of variants is not supported with Figure 4C and is unclear in Figure 4D. I'm afraid

that this claim may be overreaching, especially if compared to other methods of variant generation.

We agree that the data provided was not explained adequately. Based on the above recommendation, we reevaluated the data to make the following changes.

Line 263-274: “For successful deep mutational scanning, multiple substitution should be possible at each residue within the target for adequate sampling of the sequence space. In addition to adequate number of substitutions, there should be an adequate number of counts associated with each substitution for efficient tracking using deep mutational scanning for fitness estimates. Therefore, we next analyzed the average codon substitutions and position wise variant frequencies for each of the targets. For each gene library, we scraped ~20,000 colonies. We observed several substitutions at each targeted residue within each target gene (**Figure 4D**), we observed 7.75, 7.5, and 7.55 mean substitutions per codon for *crp*, *mreB* and *rpoB* respectively (**Figure 4D**). The variant counts for substitutions at each position primarily varied between 10^2 – 10^4 counts per variant (from $\sim 2 \times 10^6$ total counts) with 439, 613 and 692 median reads per variant for *crp*, *mreB* and *rpoB* respectively (**Figure 4D**). Therefore, using CREPE we were able to successfully develop mutation libraries over the entire targeted sequence space with significant counts associated with variants for fitness mapping, of several genes in *E. coli*. “

We also updated **Figure 4D**.

Lines 259-260: why were these concentrations of rifampicin chosen?

We added an explanation for our choice of Rifampicin concentrations as follows:

Line 295-299: “As we discuss later, the concentration of rifampicin can impact the fitness of mutations. Therefore, we compared the change in frequency of mutants in the library immediately after construction (t_0) as well as after growth on three different concentrations of rifampicin (10 $\mu\text{g/ml}$: which is slightly lower than the MIC of 12 $\mu\text{g/ml}$, 100 $\mu\text{g/ml}$: the standard concentration for selection in the laboratory and 50 $\mu\text{g/ml}$: an intermediate between the two) as well as in the absence of Rifampicin in biological triplicates (t_f) (**Figure 5B**).”

Lines 265-266 and Figure 5D: To further support the claim that the resulting sequenced mutations after selection reflected on the diversity of the library, it is recommended that a no-drug control (also grown for the same number of doublings as the treatment samples) be included to account for non-library mutation selection.

This was a great suggestion made by the reviewer. In a response to this comment, we added a no drug control to our analyses. We updated the following text:

Lines 295-300: “As we discuss later, the concentration of rifampicin can impact the fitness of mutations. Therefore, we compared the change in frequency of mutants in the library immediately after construction (t_0) as well as after growth on three different concentrations of rifampicin (10 $\mu\text{g/ml}$: which is slightly lower than the MIC of 12 $\mu\text{g/ml}$, 100 $\mu\text{g/ml}$: the standard concentration for selection in the

laboratory and 50 µg/ml: an intermediate between the two) as well as in the absence of Rifampicin in biological triplicates (tr) (**Figure 5B**).”

Line 305-307: “In comparison, such significant peaks in mutation frequency were not observed in the no-drug control, where the library was grown overnight (for the same number of doublings as the Rifampicin plates) on LB plates without any Rifampicin (**Figure 5D**).”

We also mapped the distribution of fitness effects for the no-drug control and added an Appendix Figure 5:

Line 328 to 329: “In comparison, we observed that the peak for distribution of fitness effects for most mutations were centered around 0 in the absence of Rifampicin (**Appendix Figure 5**).”

Line 344 (paragraph): the findings are interesting and would benefit from restructuring of the paragraph. An initial comment on a "trade-off" between the potential detrimental effects of mutating of an essential gene vs. the fitness advantage of mutation for drug resistance would be helpful here.

We restructured our paragraph as per the reviewer’s recommendation:

Line 386-403: “At times variants with lower MICs are preferentially selected at lower rifampicin concentrations in the laboratory and clinic (Lindsey et al. 2013; van Ingen et al. 2011; Berrada et al. 2016). Mutations with high rifampicin resistance confer growth defect because they occur close to the catalytic site of RNA polymerase (Campbell et al. 2001) and consequently are detrimental to the cell (Brandis and Hughes 2018). The prevalence of some lower MIC variants could be due to a trade-off between resistance and detrimental effects of mutations. Interestingly, at the rifampicin concentration of 10 µg/ml the maximum fitness for some mutations resistant to only 10 µg/ml was comparable to the maximum fitness for mutations selected at 50 and 100 µg/ml (**Figure 6D** and **Figure EV 3A**). These variants selected at lower rifampicin concentration had weaker resistance and lower MICs compared to the ones selected at 50 and 100 µg/ml (**Appendix Figure 6** and **7**). We posited that the comparable fitness could be due to the above-mentioned trade-off between resistance and growth defects. Since the substitutions in weakly resistant mutations with a low MIC are less bulky and/or away from the active site (**Figure EV 3B** and **EV 3C**), they may have less detrimental effects. Therefore, at lower concentrations of rifampicin, the strongly resistant mutations have high fitness mainly due to their strong inhibition of rifampicin binding. However, at the same lower rifampicin concentration the weakly resistant mutations may have their fitness equivalent to the strongly resistance mutations due to a relative growth advantage, despite the weaker inhibition. This hypothesis was further confirmed when we found that the low-resistance mutations grew significantly better than the high-resistance mutations in the absence of rifampicin (Student’s t-test p-value < 0.01, **Figure 6E**).”

Line 355: how were the 8 mutants chosen for the analysis shown in EV-3D? Please consider moving this to a main figure.

We chose the 8 mutations based on their fitness at low and high Rifampicin concentrations. We moved the supplementary figure to the main text as **Figure 6E**.

Line 401-403: “This hypothesis was further confirmed when we found that the low-resistance mutations grew significantly better than the high-resistance mutations in the absence of rifampicin (Student’s t-test p-value < 0.01, **Figure 3E**).”

Can the fitness scores of additional resistant mutants be quantified from no drug pools?

The comparison of no-drug pool mutations is a great idea. We performed the comparison and we saw that the fitness of resistant mutations was significantly lower than the non-resistant mutations in the absence of rifampicin:

However, in the absence of Rifampicin there is no selective pressure. Consequently, the diversity in the population after growth without Rifampicin is very high. Therefore, in order to perform accurate fitness estimates in the no-drug populations, we need to get a significantly higher number of reads to access rarer mutations. Using the data, we got for the no drug control, we were able to score only 20 resistant mutations amongst the 40 identified. This could be because the lack of selection also decreases the quality of fitness estimates for deleterious mutations.

However, as a future study we want to perform competition assays with our selected pool of mutations in liquid media without any drug and at different concentrations of Rifampicin. In this experiment the population would be enriched with resistant mutations. We will also sample the mutations at several time points. Both higher diversity and a time course will help us get better fitness estimates to characterize the mutations. However, this study is beyond the scope of the current paper as the major highlight of the current study is the genome modification technology.

Epistasis analysis and Figure 7: How were the shown mutants selected? In Figure 7A, are the epistatic relationships statistically significant? The fraction of mutations with epistatic relationships is exceptionally high and would benefit from a thorough statistical analysis. Figure 7C and 7D do not add significant information - please consider removing.

We agree with the reviewer. We were surprised to find such a high fraction of epistatic relationships in our data as well. However, we think that the high fraction is likely due to the strong antibiotic selection. Due to the selection, we are looking at a very niche population of resistant mutations. We did perform statistical analysis to classify variants. The details can be found in the methods section.

We removed figure 7C and 7D.

Minor comments:

Lines 185-187: Refer to figure 2E to illustrate that expression of MutL-E32K increased the maximum mutation frequency.

We added the reference

Lines 337-338: consider rewording

We reworded the text to improve clarity:

Line 379-382: “This demonstrates that resistance mechanisms for mutation at certain residues such as S531 and I572 are more complex than just steric inhibition. Our results corroborate a recent finding that high resistance is an outcome of several changes within the binding pocket (Molodtsov et al. 2017).”

Line 702: remove first "to"

We removed the “to”.

Line 745-747: Repeated from previous figure description.

We removed the repetition.

Line 560: Kapa (not Kappa)

We updated the polymerase name.

Lines 562-564: consider rewording sentence (Paired end 2*300bp read sequencing was performed using the Miseq platform).

We reworded it.

Line 722:724: “Paired end 2*300bp read sequencing was performed using the Miseq platform following manufacturer guidelines for sequencing.”

Figure 5E: description of black line missing

We added the description:

Line 877-879: “salmon with the black line representing a fit for the normal distribution to estimate the mean and standard deviation of the distribution of fitness effects”

Reviewer #3:

The authors present somewhat incremental advances in recombineering-based library construction at endogenous loci of *E. coli*, with a particular focus on conducting "deep mutational scans" of essential genes. While the tweaks that improved the experiment technically were previously published (e.g. increasing homology arm length, dominant negative inhibition of *mutL*), the authors offer, in addition to their method, an informative, high-quality dataset examining the effects of many mutations in two regions of *rpoB*. Thus, the method seems like a valuable tool for the mutation of essential genes in *E. coli*. The target audience of the advances presented in the manuscript may be limited to those focused on genome engineering in *E. coli*; while this fact dampens my own enthusiasm, the work's overall focus on a method to improve genome engineering seems appropriate for MSB.

Major concerns

My understanding of a deep mutational scan is an experiment that characterizes the majority of possible beneficial, neutral, and deleterious mutants with equal depth. However, in the cases presented here, many mutations integrated at the essential gene are immediately lost in the population due to their deleterious effects on gene function. This loss accounts for the reduced "editing efficiency" at essential vs. non-essential genes (~50% vs 80%). It would be useful for the authors to characterize those variants that are integrated and subsequently lost as lethal based on their frequency in the donor library. The input frequency of each variant should allow the experimenters to define an expectation that conservatively separates chance drop-outs from genuinely deleterious variants. I generally agree with the statement in the methods that within "an essential gene, the occurrence of stop codons would be impossible," but these stop-containing variants are a useful reference point in deep mutational scans, even in an essential gene.

We completely agree with the reviewer on the above comment. The above comment made by the reviewer is extremely insightful. As a matter of fact, we did compare the frequency of mutations between the donor and the genome (See figure below). We demonstrate the data for *galK* and *rpoB*. In each plot we stop the distribution of fitness effects for all mutations, stop codons and synonymous mutations. In *galK* we observed that the distribution of fitness effects for all mutations, stop codons and synonymous mutations overlapped with each other. In addition, as you can observe, the overall fitness for all mutations was negative. The negative fitness was expected because the wild-type reference in our analysis was the sequence with only the PAM mutation. As we have discussed in the paper, there is a strong preference for incorporation of the template with the PAM mutation only (**Figure 2 and 3**). So comparatively, the integration frequency of sequences with mutations in addition to PAM are always lower than the SPM, which explains the negative fitness.

However, in stark contrast, we observe that the distribution of fitness of mutations in *rpoB* has a peak only slightly lower than 0. However, the distribution has a long tail with several mutations with very high negative fitness of integration. As recommended by the reviewer, we used stop codons as a reference, and added the synonymous mutations to the analyses. We observe that the distributions for

synonymous and stop mutations do not overlap (**Figure B**). As expected, the stop codons have a negative fitness (**Figure B**). The peak for all mutations overlaps with the peak for the synonymous mutations. Therefore, in the case of essential genes there is a clear signal that can help differentiate mutations that are truly deleterious to the gene.

We used the stop codons as a reference to identify such mutations. We were able to identify 24 mutations that could be potentially deleterious. We also used the synonymous mutation peak to classify several mutations as near neutral. Therefore, based on the reviewer's remarks, we can use CREPE to potentially identify very deleterious mutations in essential genes. However, we did not include the data in the paper because we are now trying to identify methods which we can use to characterize these deleterious mutations. We did use a homology based DCA model to predict mutation scores (higher scores correspond to more detrimental mutations). We do observe that the deleterious mutations had a slightly significantly higher score as compared to the neutral mutations (p-value = 0.011, Figure above panel C). However, the correlation between the mutation score and the genome integration fitness was very poor. We are currently working on developing other methods to validate such deleterious mutations.

That said, the above findings do not discount the fact that using CREPE we were able to isolate several deleterious but non-lethal mutations. In order to highlight the importance of such mutations, we discuss the section on fitness cost of Rifampicin resistance (Figure 6). Also, such mutations can be observed in our no-rifampicin control (Appendix Figure 4). However, an in depth evaluation of these

mutations is beyond the scope of this paper due to further requirement of validation tools.

Related to the above comment, stop codons are used to filter the data, but the reader is not provided with a representation of how deleterious the stop codons are in these data and these data ought to be shown. I generally agree with the statement in the methods that within "an essential gene, the occurrence of stop codons would be impossible," but these stop-containing variants are a useful reference point in deep mutational scans, even in an essential gene.

In order to address the reviewer's comments, we updated the section centered around figure 4 and now represent the fraction of reads with synonymous, non-synonymous and stop codon mutations before and after the integration on the genome. Please read the updated section below:

Line 239-262: "For each of the target genes, we observed a high mutation frequency per base across the targeted region (**Figure 4A**). However, the mutation efficiency for these targets was significantly lower than *galK* (**Figure 4A**). We observed a higher percentage of variants with only the synonymous PAM mutation in the donor libraries for these targets, which could explain the lower mutation efficiency in these targets compared to *galK* (**Figure 4A**). We used the same error-prone PCR protocol for each of these targets as *galK* (**Figure 4B**). Since the mutation frequency using error-prone PCR is directly proportional to amplicon length, the mutation efficiency was lower in the donors for the new targets compared to *galK*.

However, between these targets, the mutation efficiency for *rpoB* and *mreB* was significantly lower than that of *crp*, even though they had the same target size. The lower editing could be due to the essentiality of *rpoB* and *mreB*. Therefore, deleterious mutations would not be tolerated on the genome. We expected that sequences with higher number of mutations would deplete for essential genes because a higher number of mutations are more likely to be deleterious. Therefore, we compared the distribution of variants based on number of mutations per sequence (**Figure 4B**). While the distribution was comparable in the donor, we observed fewer sequences with greater than two mutations in addition to the PAM in *rpoB* and *mreB* compared to *crp* (**Figure 4B**). To further validate our hypothesis, we also compared the percentage type of mutation (synonymous, non-synonymous and stop codons) between the donor and the genome. For *rpoB* and *mreB* the percentage of synonymous substitutions increased, the percentage of non-synonymous substitutions decreased, and substitutions leading to stop codons diminished (**Figure 4C**). This observation is expected as non-synonymous mutations are more likely to have a functional impact than synonymous mutations and stop codons in essential genes would be deleterious. Few sequences with stop codons occurred in *rpoB* and *mreB* likely due to sequencing and PCR errors (**Figure 4C**). Therefore, although we obtained high mutation efficiency for essential genes, the essentiality of these genes impacted the spectrum and distribution of mutations."

Is there a reason that mutagenesis data from the other scanned genes are not presented? *galK*, *crp*, and *mreB* are all presented in Figure 4A, but are not revisited. A clearer explanation as to why *rpoB* is the focus would be helpful

We agree with the reviewer. A reason for the choice of our target genes was their importance in laboratory evolution. However, the fitness improvements due to mutations in *mreB* and *crp* are due to global cellular changes that may be difficult to interpret. In comparison, *rpoB* was a better candidate for a quick demonstration of fitness estimates as it had a clean selectable phenotype (rifampicin resistance), which has been extensively studied over the past three decades. As a justification we added the following sentence in the text:

Line 278-282: “Although each of our target regulatory and essential genes have been found to play an important role in evolution, we chose to focus the validation of fitness estimates on *rpoB* for two reasons. Firstly, resistance to antibiotics has a clean survival/non-survival outcome that is easily interpretable. Secondly, resistance to Rifampicin has been extensively characterized over the past three decades which allowed adequate interpretation of our fitness estimates.”

along with at least a minimal characterization of the data of the other three genes.

In order to answer the comment, we now demonstrate the frequency of codon substitutions as mentioned below. We also now describe the impact of different codon substitutions as highlighted in the previous comment. All revisions are updated in Figure 4. In addition to the details about synonymous, non-synonymous and stop codons discussion above, we added the following description:

Line 263-274: “For successful deep mutational scanning, multiple substitution should be possible at each residue within the target for adequate sampling of the sequence space. In addition to adequate number of substitutions, there should be an adequate number of counts associated with each substitution for efficient accurate fitness estimates. Therefore, we next analyzed the average codon substitutions and position wise variant frequencies for each of the targets. For each gene library, we scraped ~20,000 colonies. We observed several substitutions at each targeted residue within each target gene (**Figure 4D**), we observed 7.75, 7.5, and 7.55 mean substitutions per codon for *crp*, *mreB* and *rpoB* respectively (**Figure 4D**). The variant counts for substitutions at each position primarily varied between 10^2 – 10^4 counts per variant (from $\sim 2 \times 10^6$ total counts) with 439, 613 and 692 median reads per variant for *crp*, *mreB* and *rpoB* respectively (**Figure 4D**). Therefore, using CREPE we were able to successfully develop mutation libraries over the entire targeted sequence space with significant counts associated with variants for fitness mapping, of several genes in *E. coli*.”

The section "Alternate RecA-mediated mechanism may explain recombination using templates with single nucleotide changes" feels out of place in the Results section. The experiments in the paper may offer insight into the mechanism of action, but this mechanism is not presented clearly or convincingly enough to add weight to the rest of the paper's results. I suggest removing this section.

We removed this section and have an abridged version of the model now in the discussion instead. We have removed the model figure as well.

Minor concerns

The size of the window for efficient mutagenesis is presented variably throughout

the paper. The effective size in Figure 4 of ~200 bp seems significantly lower than the suggestion in the discussion that 300-400 bp regions are easily targeted by the method.

We demonstrate the 300-400 bp regions are easily targeted using *galK* as a reference. We reduced the size to 250 bp for the other genes because we wanted to have the entire gene within one illumina read with over 100% overlap (2X300 Miseq), to improve the read quality. We added a justification in our results section: Lines 236-239: “We targeted a shorter window for these essential genes compared to *galK* because with the shorter window size, the entire targeted region can be covered by a single read using a paired-end 2*300Miseq. The complete overlap between the forward and reverse reads allows for improved read quality and reduces error.”

A window of ~250 bp was targeted for mutagenesis, but no explanation is offered for the positional variability in mutagenesis rate outside of the PAM-adjacent bias, in particular the steep drop-off near the borders.

We targeted a 230 bp long window. 20 bp regions in the ends on each side are not mutated. Therefore, we observe a steep drop-off in these regions. In the previous version of the paper we reported the window size as 270 including these unmutated regions. We apologize for the same. We have now updated the text to highlight that the window size is 230 bp:

Lines 236-239: “We targeted 230 bp long regions in the genes: *crp*: a non-essential gene encoding a global metabolism regulator that controls the expression of hundreds of proteins in *E. coli* (Görke and Stülke 2008), *rpoB*: an essential gene that encodes the beta subunit of RNA polymerase, and *mreB*: an essential gene that encodes a cytoskeletal protein (**Figure 4A**).”

A more useful axis would be valuable in Figures 4C. It would be more helpful to show the number of mutations per codon rather than the more abstract frequency.

We updated figure 4 to now include the codon change data.

Also with the target size, the paper's introduction and discussion extol the benefits of using a single, efficient gRNA, but undercut that benefit by suggesting that guides tiling across a gene will be necessary for doing more comprehensive deep mutational scans. Because this is mostly a methods paper, the authors should make a clearer case for the utility of each approach, rather than casting the CREATE and CREPE methods as hero or villain on demand.

We updated the text to clearly delineate the limitations now:

Line 535-539: “In its current format, there are several limitations of using CREPE. Firstly, the technology is the limited target size (300-400 bp). In order to overcome this limitation, we will have to use a tiling approach to study entire proteins using shorter windows. We will need to identify functional gRNA for each tile. Therefore, studies using CREPE was limited to only particular genomic regions. For genome wide studies previous high-throughput and multiplexed platforms are still the only viable approach.”

In the discussion of the rifampicin resistant mutants, the text does not comment on

an important class of mutations: those resistance mutations known to the literature and that were tested in this assay but showed no resistance. If there are none in this class, it would be worth mentioning. Information appears to be in Appendix Table 1.

Yes, there are mutations that are known to be resistant to Rifampicin and not found here. A major reason for the same is codon usage. Some resistant amino acid substitutions are two nucleotide changes away which are impossible to achieve by error-prone PCR. We added that in our text:

Line 539-549: “Secondly, using error prone PCR to generate libraries only generates single nucleotide changes per codon. Therefore, for each amino acid a maximum of 11 other amino acid substitutions are accessible. Several previously known mutations such as the clinically relevant mutation S531L were not identified in our study because they require two consecutive nucleotide substitutions. This mutation has been identified in *Mycobacterium tuberculosis*, which has a different codon usage than *E. coli*. Therefore, while this mutation is accessible by a single nucleotide change in *M. tuberculosis*, it would require two consecutive nucleotide changes in *E. coli*. Such changes are hard to achieve using error-prone PCR.”

In the discussion of epistasis, Figures 7A and 7C appear use different thresholds for identifying instances of negative or positive epistasis. Please clarify in the text.

From the text, it appears that negative epistasis occurs when "actual fitness was lower than the fitness-sum" (lines 393-394). Does this mean that the threshold was simply any value in which the deviation from the fitness-sum was < 0 ? The methods section indicates that some statistical cutoff was used, but this isn't apparent in the results text or figure legend.

In response to both of the above comments, we added a short description of the epistasis measurements as following:

Line 425-429: “Epistasis occurs when the fitness of mutation combinations deviates from 0. However, there are errors associated with fitness estimates using deep sequencing (**materials and methods**). Therefore, we assigned epistasis signs to the mutation combinations when whole fitness deviated from the sum of mutations after accounting for the error in the fitness estimates for individual mutations (**materials and methods**).”

I was confused by the statement on lines 453-454 that "using error prone PCR to generate libraries only generates single nucleotide changes." This study contains plenty of templates that had more than a single nucleotide change, but I believe the authors refer to consecutive mutation events on a single template. This limitation would prevent access to certain codons that require more than a single nucleotide change.

We updated this section as follows to better clarify our argument:

Line 535-549: “In its current format, there are several limitations of using CREPE. Firstly, the technology is the limited target size (300-400 bp). In order to overcome this limitation, we will have to use a tiling approach to study entire proteins using shorter windows. We will need to identify functional gRNA for each tile. Therefore, studies using CREPE was limited to only particular genomic regions. For genome wide studies previous high-throughput and multiplexed platforms are

still the only viable approach. Secondly, using error prone PCR to generate libraries only generates single nucleotide changes per codon. Therefore, for each amino acid a maximum of 11 other amino acid substitutions are accessible. Several previously known mutations such as the clinically relevant mutation S531L were not identified in our study because they require two consecutive nucleotide substitutions. This mutation has been identified in *Mycobacterium tuberculosis*, which has a different codon usage than *E. coli*. Therefore, while this mutation is accessible by a single nucleotide change in *M. tuberculosis*, it would require two consecutive nucleotide changes in *E. coli*. Such changes are hard to achieve using error-prone PCR. Additionally, previous studies have highlighted that 2-3 consequent nucleotide changes, to enable complete saturation mutagenesis to more distant amino acids, often lead to more diverse chemical changes that have larger fitness effects (Garst et al. 2017; Pines et al. 2015).”

2nd Editorial Decision

10th February 2020

Thank you again for sending us your revised manuscript. We have now heard back from the two referees who were asked to evaluate your study. As you will see below, the reviewers acknowledge that the study has improved as a result of the performed revisions and think that it is now suitable for publication. Reviewer #3 recommends adding to the manuscript (e.g. as an EV or Appendix Figure) the figure that is currently only included in your point by point response to their comments.

REFEREE REPORTS

Reviewer #1:

The authors have done a good job at addressing our comments.

Reviewer #3:

The authors have adequately addressed my concerns, as well as those from other reviewers.

My only reservations come from the instances in the point-by-point where the authors have opted to respond with an analysis that they say will not be included in the final manuscript. See comment in point-by-point "Therefore, based on the reviewer's remarks, we can use CREPE to potentially identify very deleterious mutations in essential genes. However, we did not include the data in the paper because we are now trying to identify methods which we can use to characterize these deleterious mutations." Some analyses may be beyond the scope of this work (building predictive models or doing further validation of mutants), but some (like showing the distribution of stop codons, synonymous mutations, and missense mutations in *rpoB*) are standard in the field.

That said, I think the authors have put great effort into this work and its revision; the clarity of the manuscript has significantly increased. Their improved method will be useful in the bacterial deep mutational scanning field.

Response to Reviewers:**Reviewer #1:**

The authors have done a good job at addressing our comments.

We thank the reviewer for taking the time to review our work. We are pleased to know that the reviewer thinks that the work merits publication.

Reviewer #3:

The authors have adequately addressed my concerns, as well as those from other reviewers.

My only reservations come from the instances in the point-by-point where the authors have opted to respond with an analysis that they say will not be included in the final manuscript. See comment in point-by-point "Therefore, based on the reviewer's remarks, we can use CREPE to potentially identify very deleterious mutations in essential genes. However, we did not include the data in the paper because we are now trying to identify methods which we can use to characterize these deleterious mutations." Some analyses may be beyond the scope of this work (building predictive models or doing further validation of mutants), but some (like showing the distribution of stop codons, synonymous mutations, and missense mutations in *rpoB*) are standard in the field.

That said, I think the authors have put great effort into this work and its revision; the clarity of the manuscript has significantly increased. Their improved method will be useful in the bacterial deep mutational scanning field.

In order to address the reviewer's comments, we have added the data for deleterious mutations as Extended Figure 2 and Appendix Table S1. Please see the updated text that we added in **Page 11-12: lines 276-296**

“CREPE can be used to identify potentially deleterious mutations in essential genes

Since, we observed a significant depletion of stop codons (loss of function mutations) for essential genes (**Figure 4C**), we posited that we could compare the change in frequency of variants between the donor library and after integration on the genome to identify deleterious mutations for essential genes in *E. coli* (**Figure EV2A**). We measured fitness of substitution as log change in frequency between the donor library and the frequency after integration on the genome (**Figure EV2B**). We compared the distribution of fitness after integration for non-synonymous, synonymous and stop codon substitutions for the non-essential *galK* gene and the essential *rpoB* gene (**Figure EV2A**). In *galK*, we observed that the distribution of fitness effects for non-synonymous, stop codons and synonymous substitutions overlapped with each other (**Figure EV2B**). This was expected as the nature of substitution in a non-essential gene should not impact cell survival after integration and consequently fitness. However, for *rpoB*, in stark contrast, the

distributions for synonymous and stop mutations did not overlap (**Figure EV2B**). The distribution of fitness for synonymous mutations was centered slightly below 0 and that for the stop codons centered around -4 (**Figure EV2B**). Therefore, in the case of essential genes we observed a clear signal to differentiate between deleterious and non-deleterious mutations in the gene (**Figure EV2B**). The distribution of fitness of non-synonymous mutations in *rpoB* had a peak overlapping with the synonymous mutations and a long tail with several mutations with high negative fitness that was comparable to the fitness of the stop codons (**Figure EV2B**). We used the distribution of fitness for stop codons as a reference to find 25 substitutions that could be potentially deleterious in *rpoB* (**Appendix Table S1**). Therefore, we can use the CREPE technology to identify potentially deleterious substitutions in essential genes in *E. coli*. Identification of such substitutions could be important for identification of functional residues in poorly characterized essential genes. ”

We thank the reviewer for taking the time to review our work. We are pleased to know that the reviewer thinks that the work merits publication.

Accepted

13th February 2020

Thank you again for sending us your revised manuscript. We are now satisfied with the modifications made and I am pleased to inform you that your paper has been accepted for publication.

Corresponding Author Name: Ryan Gill and Olivier Tenaille

Manuscript Number: MSB-19-9265